# Temporal genomics in Hawaiian crickets reveals compensatory intragenomic coadaptation during adaptive evolution

Xiao Zhang [1,2] ✉, Mark Blaxter [3], Jonathan M. D. Wood [3], Alan Tracey[3], Shane McCarthy [3], Peter Thorpe[4,7], Jack G. Rayner[2], Shangzhe Zhang[2], Kirstin L. Sikkink[5], Susan L. Balenger[6] & Nathan W. Bailey [2] ✉

Theory predicts that compensatory genetic changes reduce negative indirect effects of selected variants during adaptive evolution, but evidence is scarce. Here, we test this in a wild population of Hawaiian crickets using temporal genomics and a high-quality chromosome-level cricket genome. In this population, a mutation, *flatwing*, silences males and rapidly spread due to an acoustically-orienting parasitoid. Our sampling spanned a social transition during which *flatwing* fixed and the population went silent. We find long-range linkage disequilibrium around the putative *flatwing* locus was maintained over time, and hitchhiking genes had functions related to negative *flatwing*-associated effects. We develop a combinatorial enrichment approach using transcriptome data to test for compensatory, intragenomic coevolution. Temporal changes in genomic selection were distributed genome-wide and functionally associated with the population's transition to silence, particularly behavioural responses to silent environments. Our results demonstrate how 'adaptation begets adaptation'; changes to the sociogenetic environment accompanying rapid trait evolution can generate selection provoking further, compensatory adaptation.

One of the controversial legacies of the Modern Synthesis is the idea that allele frequency changes directly caused by natural selection should provoke compensatory evolution at other loci[1]. This could happen in two ways. First, adaptive loci can alter the genetic environment through pleiotropy and epistasis, exerting selection on other regions of the genome to minimise negative genetic interactions and maintain organismal functioning[2,3]. This outcome was described as "coadaptation" by Dobzhansky[4], who also emphasised that the genotypes contained within a local population constitute a system to which individuals adapt. Second, the spread of adaptive variants through populations can cause indirect selection to mitigate negative

consequences of altered social or ecological interactions caused by the variants themselves, a process of rapid modification that Fisher described nearly a century ago[5]. More recently, a "selection, pleiotropy and compensation" model that integrates both mechanisms to derive genetic expectations was developed by Pavlicev and Wagner[6], who argue that most signatures of genomic adaptation are attributable to compensatory responses, rather than direct responses to external selection.

The steps by which compensatory, intragenomic coadaptation occurs have remained under-studied despite their importance in determining the course of adaptive evolution. Abundant theory

[1]Tianjin Key Laboratory of Conservation and Utilization of Animal Diversity, College of Life Sciences, Tianjin Normal University, Tianjin, China. [2]Centre for Biological Diversity, School of Biology, University of St Andrews, St Andrews, Fife, UK. [3]Tree of Life, Wellcome Sanger Institute, Cambridge, UK. [4]School of Medicine, University of St Andrews, St Andrews, Fife, UK. [5]Arima Genomics, Carlsbad, CA, USA. [6]College of Biological Sciences, University of Minnesota, Saint Paul, MN, USA. [7]Present address: Data Analysis Group, Division of Computational Biology, School of Life Sciences, University of Dundee, Dundee, UK. ✉e-mail: xz42@st-andrews.ac.uk; nwb3@st-andrews.ac.uk

predicts negative pleiotropy and epistasis during bouts of adaptation, and ample evidence supports it (e.g.[6–20]). Recent theory has also sought to model genetic feedback arising from the social environment as underlying gene frequencies shift, for example through indirect genetic effects[21] or variation in social behaviour[22]. All of these lines of argument predict that compensatory evolutionary responses should be widely observable[6]. However, few empirical studies characterise genome-wide consequences of either form of feedback in wild systems, instead focusing on more tractable systems of experimental evolution (e.g.[11,23]). This is likely due to the difficulty of tracking the earliest stages of adaptive evolution in nature when relatively stronger background interactions involving adaptive loci are expected. Most bouts of rapid adaptation do not occur within the duration of a standard evolutionary study, which prevents comparison of genomic architectures before and after adaptation. Nevertheless, there are clear predictions about the outcome of evolutionary feedback arising from the rapid genetic adaptation: genomic hitchhiking during selective sweeps should expose linked haplotypes to the action of selection[24–26], evolutionary compensation of negative fitness effects associated with an adaptive variant's spread should be detectable as genomic signatures of selection, and those selected regions should be associated with functions that mitigate negative effects associated with the adaptive variant[27].

We tested this using a rapidly evolving Hawaiian population of the field cricket *Teleogryllus oceanicus* located on the island of Kauai. This system enabled us to dissect the temporal genomic dynamics of adaptive evolution in the wild due to an intensive level of detail about the timing, spread, and social consequences of a genetic variant that protects male crickets from lethal attack by an acoustically-orienting parasitoid fly (*Ormia ochracea*) (Fig. 1a). The fly is endemic to North America and the date of its introduction in Hawaii is unknown, but it is known to have coexisted with crickets on Kauai for over a decade before the protective cricket variant emerged[28]. The variant, *flatwing*, segregate as a single locus on the X chromosome and erases sound-producing structures on male wings by feminising them (Fig. 1b)[28–30]. The resulting silence protects males from detection by the eavesdropping parasitoid, a significant benefit, but simultaneously imposes costs by removing the primary signalling channel involved in mate attraction, courtship, and aggression. Despite these countervailing costs, *flatwing* spread rapidly under fly selection from its initial appearance between 2001–2003[28,31].

The temporal sequence of *flatwing's* selection in the Kauai population is well-documented[28,31–33]. By 2004, *flatwing* had reached a frequency of >90% in the population[28]. The crickets breed continuously with approximately four generations per year, indicating a rapid sweep to near – but not complete – fixation in fewer than 20 generations. In subsequent years, a small number of singing males, ca. 5%, still existed within the population and silent males employed flexible satellite strategies to respond to the remaining calling males and intercept females for mating[28,31]. There is evidence that in this Kauai population particularly, both sexes showed high degrees of behavioural plasticity which would facilitate such responses, manifested as more flexible and sensitive responses to social and acoustic signals[31,34,35]. However, between 2017 and 2018, *flatwing* spread to fixation in the same population and it was then observed to be completely silent[32,33]. This social transition represents a critical threshold: the previous satellite mating tactics are by definition ineffectual when no singing males remain, yet the population continues to thrive despite the *flatwing* mutation's detrimental impacts on fitness traits unrelated to parasitoid evasion. Meanwhile, the loss of singing males in the population may be expected to intensify the effects of fly selection on flatwing males, because flatwing males from the same population have been observed to be infected by flies (N. Bailey, *pers. obs.* 2012). The flies persist in this population despite the absence of normal-wing targets and despite the lack of alternative hosts at any appreciable abundance, suggesting continued pressure on the all-silent population (ref. 31, Bailey & J. Rayner, *pers. obs.* 2022).

The *flatwing* mutation is associated with a broad range of indirect effects, that is, impacts on traits not directly under selection due to parasitoid pressure (Fig. 1a). For example, flatwing males' cuticular pheromones are feminised[36], they cannot attract mates using long-range acoustic signals but still incur the energetic expenditure of wing movements associated with singing[37,38], they have reduced short-range courtship success[39], and they fare more poorly in intrasexual aggressive encounters[40]. Females carrying *flatwing* alleles have higher rates of mating failure[41] and show slower growth and smaller reproductive tissues[42]. The experience of silence, which is a result of *flatwing's* sweep through populations, causes plastic changes in physiology and behaviour which appear to have been an important factor in the mutation's rapid fixation. In populations dominated by flatwing males, both sexes lack information about the availability and location of sexually mature males. Females become more responsive and less discriminating of mates to compensate for this[34,43], males are more likely to adopt satellite mating tactics[44], and both sexes show greater locomotion[35,45].

To test whether and how these consequences exert selection to increase genomic coadaptation, we first generated a high-quality chromosome-contiguous reference genome for *T. oceanicus*. With this, we tested for temporal patterns of genomic selection on gene sets relevant to such compensatory responses. The procedure involved resequencing male crickets collected from the Kauai population over three-time points to detect genomic signatures of selection, then overlapping these with genes found to be differentially expressed in Kauai crickets. The latter individuals had been exposed to treatments mimicking selection imposed by the parasitoid fly and by the altered social environment caused by *flatwing's* spread. Supplementary Fig. 1 provides a diagram of the workflow. Our analyses focused on the population after flatwing crickets had invaded and spread, with temporal sampling spanning the transition from when flatwing males comprised ca. 95% of the population to 100% fixation (Fig. 1b). This sampling scheme enabled us to identify genomic selection associated with compensation to negative pleiotropic effects of *flatwing* as well as to the conspicuous shift in the social environment (silence) brought about by *flatwing's* fixation (Fig. 1a). Our analyses revealed genomic consequences of the population's transition to silence; selected regions were enriched for biological functions relevant for mitigating negative consequences of evolved song loss. Given the importance of cricket immunological responses to invasion by parasitoid larvae and behavioural strategies for obtaining mates depending on whether singing males are present or not, we investigated genomic responses to changed selection at immunological and behavioural levels using candidate gene sets derived from gene expression data. Characterising the temporal dynamics of selection on immune-related genes was performed as a useful contrast because fly selection pressure was present throughout the duration of the sampling interval. In this way, we identified genomic regions under selection during social transitions which are associated with unique physical and behavioural phenotypes observed in silent environments. Social and genetic changes caused by *flatwing* and its rapid spread spawned evolutionary feedback which had wide-reaching, almost immediate, impacts consistent with intragenomic coevolution.

## Results

### Temporal population genomic dynamics of rapidly evolving wild crickets

We studied the population of *Teleogryllus oceanicus* in which flatwing male crickets were originally discovered on the Hawaiian island of Kauai[28]. By resequencing individuals from this population sampled at different timepoints, we were able to characterise the temporal dynamics of a single population with independently verified information about key demographics and sources of selection. To enable

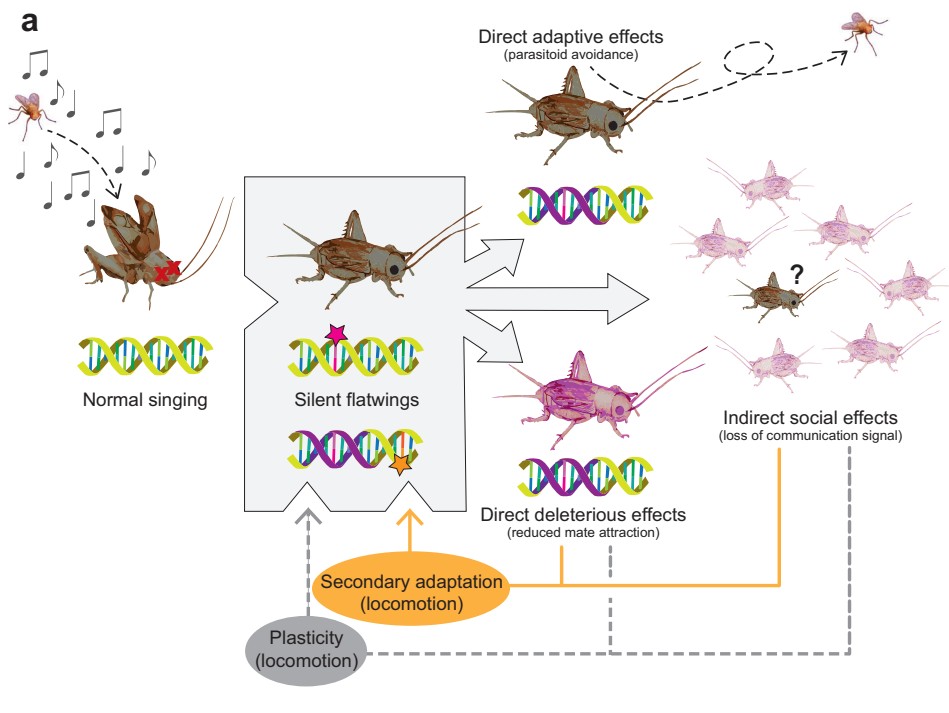

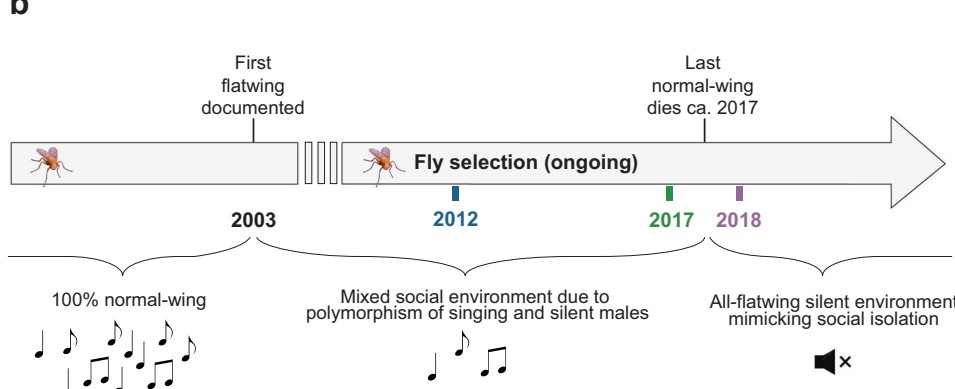

**Fig. 1 | Conceptual diagram describing compensatory genomic coadaptation in silent Hawaiian crickets. a** Feedback loops between an adaptive variant under selection, *flatwing*, and genome architecture. A population containing only normal singing *Teleogryllus oceanicus* males is subject to strong selection from lethal, acoustically-orienting parasitoid, *Ormia ochracea* (left). *Flatwing* invades the genome and quickly spreads under selection (middle shaded box; pink star represents the *flatwing* genomic variant). The spread of *flatwing* has phenotypic effects that are directly selected due to the adaptive benefit of parasitoid avoidance (right, top arrow). However, *flatwing* also imposes direct fitness costs via pleiotropic or other associated effects, for example through a diminished ability to attract mates (right, bottom arrow, represented by pink-coloured cricket. Deleterious effects might also arise due to genomic hitchhiking, represented by the purple-shaded DNA strands. See Main Text for examples of direct fitness costs.) The presence of *flatwing* in the population gene pool also imposes a changed regime of social selection by altering the perceived social environment (right, middle arrow, translucent pink crickets represent the inability of focal individuals to evaluate the presence of potential mates or rivals, due to disrupted communication. See Main Text for examples of indirect social effects.) Pre-existing adaptive plasticity can mitigate negative direct and indirect fitness effects of *flatwing* (bottom grey dashed arrow), but these negative consequences also generate a selection that feeds back to favour genetically-encoded secondary adaptations or epistatic allelic combinations that reduce negative effects of *flatwing* (orange arrow). Such genomic coadaptation has rarely been studied, while considerable research attention has been directed towards understanding the former process. **b** Timeline of key social environment transitions in the Kauai cricket population (based on information from[28,29,31,32,152,153]. Colour-coding of years matches that used in figures throughout this manuscript.

reliable downstream analyses and interpretation[46], we produced a new chromosome-level de novo assembly for *T. oceanicus* (*T. oceanicus* v.2) using 109 Gb (ca. 55x) Oxford Nanopore PromethION long reads from a female homozygous for the *normal-wing* genotype, from laboratory stock originally derived from the Kauai population. We polished this with 83 Gb (ca. 42x) Illumina short reads and anchored it with 242 Gb of Illumina paired-end reads from a high-throughput chromosome conformation capture (Hi-C) library. The final assembly was ca. 2.03 Gb, of which 98% was anchored to 14 pseudochromosomes with a scaffold N50 of 137 Mb (Fig. 2a, Supplementary Fig. 2a and

Supplementary Table 1). The assembly's high contig N50 (5.7 Mb) permits a more reliable assessment of long-range linkage disequilibrium and repeat content. These results, combined with the haploid chromosome number of 13 + X in *T. oceanicus*[47], suggested that the majority of the field cricket genome was successfully assembled at the chromosomal level, representing a considerable improvement for this species[36] (Supplementary Fig. 2).

Assembly statistics and comparison with other published field cricket genome assemblies, including a previous version for *T. oceanicus*[36], are provided in Supplementary Table 2. BUSCO analysis

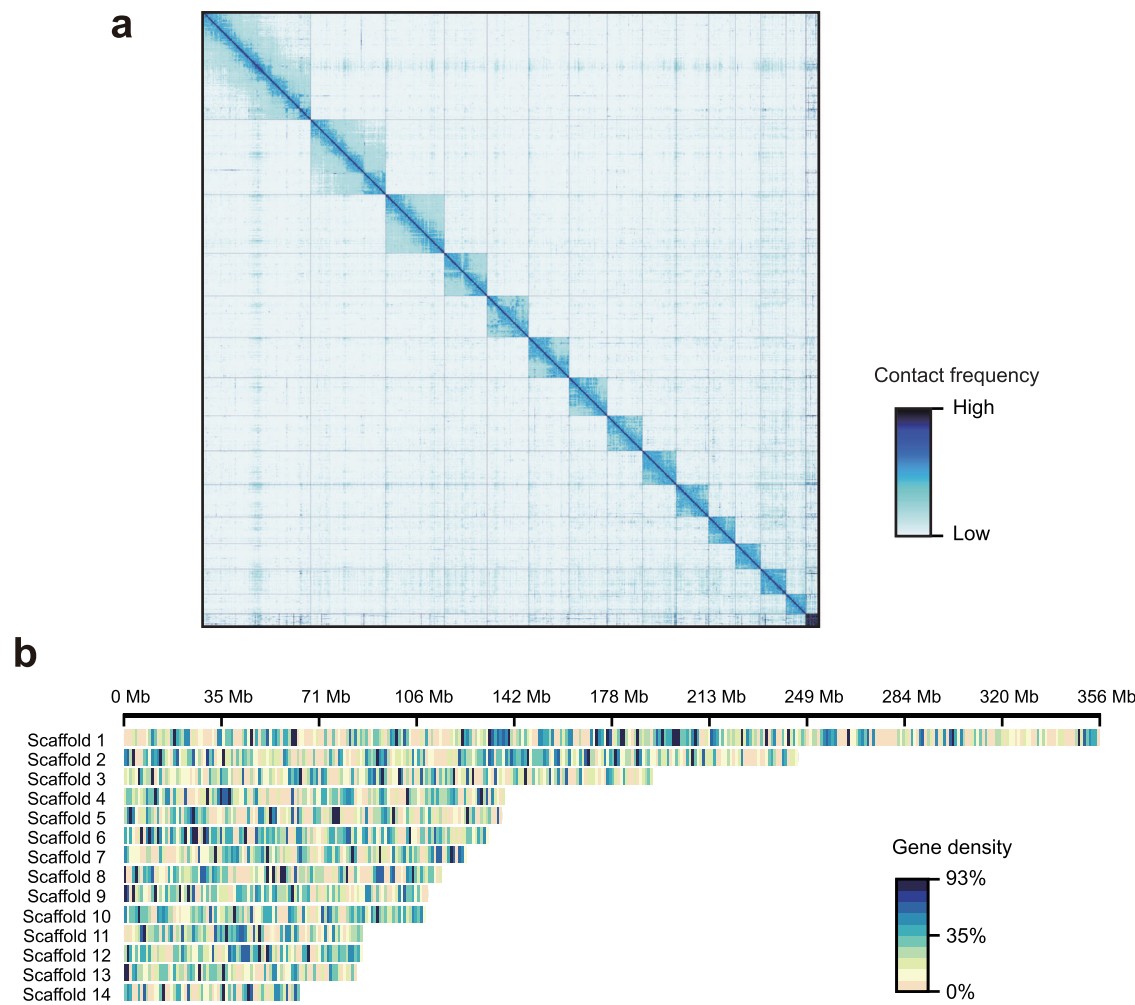

**Fig. 2 | *Teleogryllus oceanicus* chromosome-level genome assembly. a** Hi-C contact heatmap of the chromosome-level *T. oceanicus* genome assembly. **b** Genomic landscape of gene density in the *T. oceanicus* genome. Scaffold 1 was identified as the X chromosome (see 'Methods').

showed that 98% of the core Arthropoda gene set was represented (Supplementary Table 3). More than 98% of whole genome resequencing data (Supplementary Data 1, see below) and 90% of RNA-seq data (Supplementary Data 2, see below) were mapped to the assembly, confirming its high completeness. We identified 51.9% of this genome assembly as repetitive sequences (Supplementary Fig. 2c) and predicted 21,211 protein-coding genes (Fig. 2b and Supplementary Table 4). To ensure the relative completeness and accuracy of this reference gene set, we used RNA-seq data previously obtained from 114 sequencing libraries covering four tissues, three populations, both sexes and two wing morphs to aid gene prediction (Supplementary Data 2)[30,36,48–50]. In the final reference gene set, 98.8% of genes were detected in at least one RNA-seq library, 86% exhibited significant expression after strict filtration, and 77.5% recovered orthologs in public database searches, indicating a low rate of false positives and high sensitivity in our gene prediction procedure.

To interrogate the temporal population genomic dynamics of these rapidly evolving wild crickets, we sequenced and analysed the whole genomes (average sequencing depth of 28 ×; Supplementary Data 1) of individuals that had been sampled from the same population at three crucial time points: 2012, 2017, and 2018 (Fig. 1b). Australian populations are not known to experience selective pressure from the parasitoid fly, making them a valuable comparison in selection analyses and as an outgroup. Whole-genome re-sequencing (WGRS) data from 7 normal-wing Australian individuals and 10 normal-wing lab-reared Kauai individuals were used for comparison. *Teleogryllus*

*oceanicus* WGRS data from the 2017 timepoint, normal-wing individuals, and Australian individuals were previously published as part of a study examining the parallel evolution of the flatwing phenotype across three different islands of the Hawaiian archipelago[30]. For onward analyses, it was important to verify that comparisons of genomic selection across time points would not be confounded by markedly different population genomic structures. We evaluated phylogenetic relationships among the 47 samples using autosomal SNP data. A neighbour-joining (NJ) tree based on pairwise genetic distances supported the independent clustering of an Australian clade and a Hawaiian cluster (Fig. 3b). For the Hawaiian cluster, individuals were grouped into two sub-clades in accordance with wing phenotypes, which reflected founder effects among the lab-reared normal-wing samples (see also[30]).

The critical observation was that flatwing individuals collected in different years clustered together, indicating no elevated relatedness of individuals sampled in the same year which could confound analyses of genomic selection. This finding was further supported by genetic structure analysis (Fig. 3a), which supported the modelling choice of *K* = 3 groups and rejected models dividing Kauai samples into smaller groups based on sampling year (*K* = 5) (Supplementary Fig. 3). *K* = 2, which treated both wild and lab-reared Kauai individuals as a single group, was more plausible than *K* = 5, but less plausible than *K* = 3. The results of principal component analysis (PCA) based on Kauai flatwing individuals collected from the wild are also consistent with the NJ tree and structure analyses. PCs 1 ~ 9 explained a total of 15%

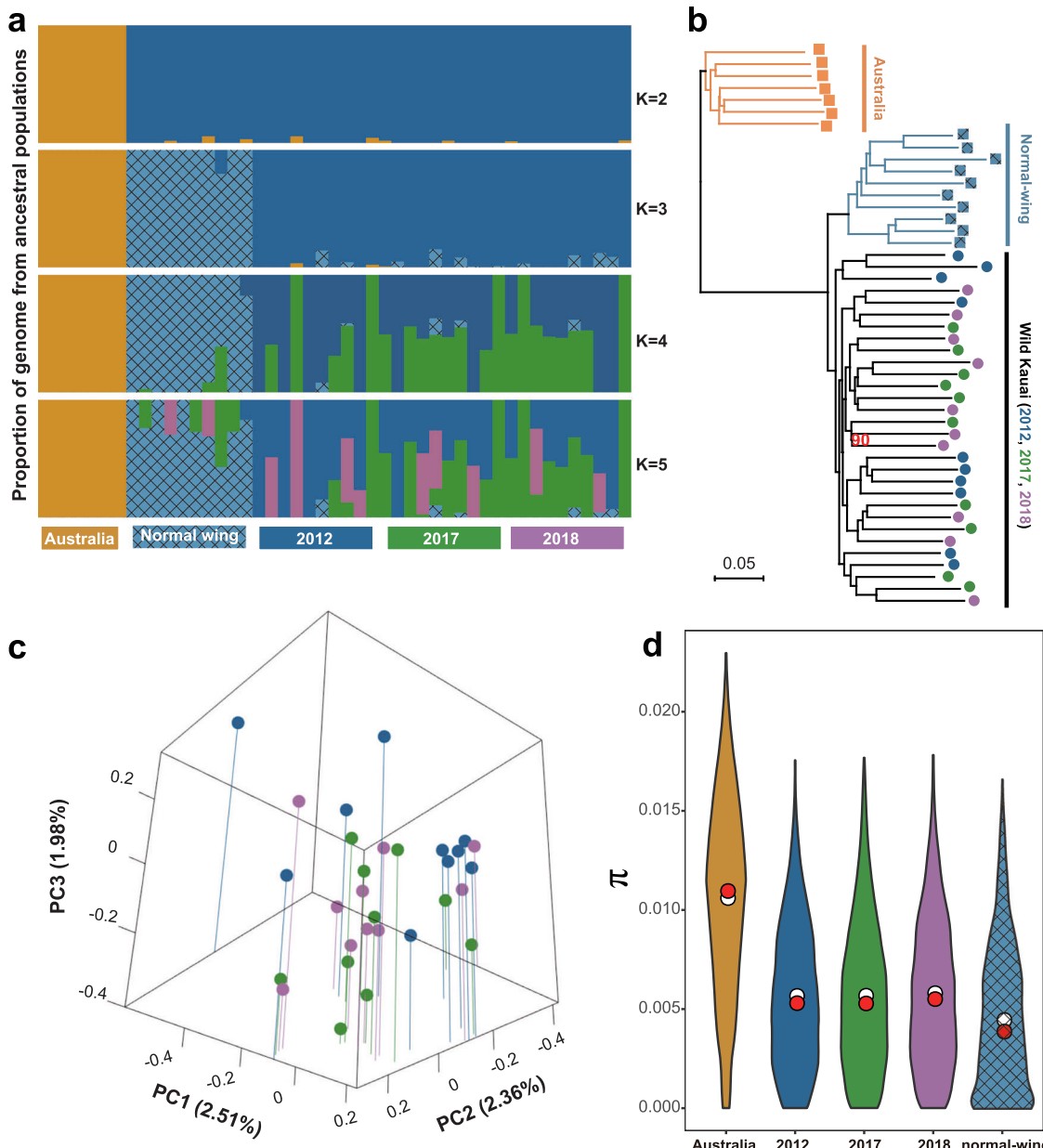

**Fig. 3 | Phylogenetic and population genetic analyses of Kauai crickets sampled at different time points. a** Population structure analysis of all individuals. The height of each coloured segment represents the proportion of the individual's genome from inferred ancestral populations. Colours reflect the scheme used in this study unless stated otherwise (desert yellow = Australian individuals, light blue with grid = lab-raised normal-wing individuals, solid blue = Kauai individuals sampled in 2012, green = Kauai individuals sampled in 2017, purple = Kauai individuals sampled in 2018). **b** Neighbour-joining phylogenetic tree based on autosomal SNPs. All nodes but one (labelled in red text) had bootstrap values of 100%. Boxes = normal-wing males, Circles = flatwing males. Colour coding as above. **c** Three-dimensional PCA scatter diagram showing genetic relationships among wild Kauai individuals based on autosomal SNPs derived from WGRS data. Vertical lines link dots to the base to aid visualisation. See Supplementary Fig. 4 for a 2-dimensional version. Source data are provided as a Source Data file. **d** Comparisons of genetic diversity (π) summarised across non-overlapping 1-Mb genomic windows for three-time points. Red and white dots represent median and mean values, respectively. Data from resequenced Australian individuals and resequenced inbred normal-wing laboratory stock originally derived from Kauai are provided for comparison.

of the genetic variation, but did not separate wild individuals collected from different years (Fig. 3c and Supplementary Figs. 4, 5). These results indicate that there is no distinct difference among the three-time points at the whole genome level, enabling us to treat all wild flatwing individuals as one population regardless of their sampling year to detect highly-resolved patterns of adaptation at the genomic level.

Mean nucleotide diversity (π) calculated using non-overlapping 10 kb sliding windows remained similarly stable across years (ANOVA, $P = 0.634$, Supplementary Table 5) (Fig. 3d), suggesting evolutionary

potential was similar across all three time periods[51–53]. Genetic diversity in all years was significantly lower in the Kauai population than the Australian population (two-sided $t$-tests: all $P < 2.2 \times 10^{-16}$), consistent with the observation that the Australian population is not under fly selection as *O. ochracea* does not exist there, it contains no flatwing males, and it has not undergone ancient bottlenecks[30]. In addition, wild flatwing Kauai individuals and wild Australian individuals showed similar Wright's inbreeding coefficients ($F = -0.011$ and $0.008$ respectively) and the inbreeding coefficient for wild flatwing Kauai samples across each sampling year remained stable (ANOVA, $P = 0.593$,

Supplementary Table 6). In contrast, the inbred laboratory line of normal-wing individuals ($F = 0.134$) showed significantly higher $F$ values than these wild flatwing individuals (Fisher's Least Significant Difference test: $P = 0.0291$, Supplementary Fig. 6 and Supplementary Table 7). These results suggest that there is no inbreeding occurring in the wild Kauai population, nor differences in inbreeding across sampling years, which is also consistent with our previous observations indicating significant gene flow shared between the Kauai population and other natural populations[30].

## Genomic consequences of *flatwing*'s rapid adaptive spread

Because the *flatwing* genotype invaded and spread in Kauai so recently, genes in its flanking regions are prone to long-range genomic hitchhiking and may be the cause of negative indirect fitness consequences of its rapid spread[30,36]. Our goal was to detect intragenomic coevolution that compensates for the indirect negative effects of *flatwing*'s fixation, both as a direct consequence to individuals carrying *flatwing*, and as an indirect consequence of its changes to the social environment (Fig. 1a). Therefore, we first assessed the extent of hitchhiking in regions flanking the candidate *flatwing* locus and related gene function in these regions to known phenotypic associations with flatwing. Second, we excluded this region from subsequent downstream analyses of intragenomic coadaptation arising from changes to the sociogenetic environment brought about by *flatwing*'s fixation to avoid confounding the two (Supplementary Fig. 1).

Previous quantitative trait locus (QTL) mapping and genome-wide association studies (GWAS) of the flatwing phenotype in Kauai localised the *flatwing* variant on 17 X-linked scaffolds dispersed across approximately one-third of the X chromosome, as well as on 27 scaffolds lacking chromosomal information[30]. A candidate 'adaptation hotspot' included the gene *doublesex* (*dsx*). *Dsx* is known to control the development of sexually dimorphic traits in insects and varies extensively in exon number, isoform number, and the presence and extent of sex-biased expression[54]. In addition to wing venation, *flatwing* also causes dramatic regulatory and phenotypic disruption such as genome-wide alteration of embryonic gene expression, feminisation of male pheromones, and increased plasticity, either via pleiotropic or genetically coupled effects[30,33,36,49,55]. As such, *flatwing*'s disruptive effects can happen at the intragenomic level, for example by changing the outcome of interactions with background genetic variation, at the individual level due to negative impacts on other traits, or at the population level by altering feedback arising from the social environment (Fig. 1a).

Here, intending to distinguish genomic signatures arising from coadaptation during social transitions from the ones caused directly by the *flatwing* genotype, we first investigated the genomic consequences of *flatwing*'s rapid adaptive spread. Taking advantage of the new chromosome-level genome assembly, we re-performed genome-wide association analyses using a larger sample of resequenced wild crickets (Supplementary Data 3) and monitored the pattern of linkage disequilibrium around the putative *flatwing* locus over time. The genome-wide association analyses revealed a single peak on the X chromosome, confirming previous findings[30,36]. It occupied approximately 20 Mb, which covered the candidate hotspot and *dsx* locus (Fig. 4a and Supplementary Data 4). Significant SNPs were directly localised within the intronic region of the *doublesex* gene, and our re-analysis of RNA-seq data using the new genome assembly confirmed a previously reported differential expression pattern of *doublesex* between flatwing and normal-wing individuals[30]. Consistent with our previous results[30], significant selective-sweep signatures covering *doublesex* were also detected using these new datasets (Supplementary Fig. 7). The goal of the current study is to test the genomic consequences of host adaptation to parasitoid flies, rather than to reveal the causative mutation(s) underlying feminised wing venation in silent crickets. We, therefore, compared haplotypes of the putative *flatwing*

locus and its flanking regions across time points in the Kauai population and examined patterns of linkage disequilibrium (Fig. 4b–e). As a control, similar analyses were also conducted on the Australian group in which neither parasitoid flies nor flatwing crickets were present (Fig. 4b, f). Our analysis revealed remarkably extensive and long-range linkage at the putative *flatwing* locus among all individuals from Kauai, irrespective of the sampling year (Fig. 4b–f). The region spanned approximately 20 Mb, included several breakpoints, and the linkage persisted for over 8 years (> ca. 32 generations), consistent with predictions for a region experiencing a strong selective sweep[24,26]. To exclude the possibility that this pattern arose due to generally high LD genome-wide, we randomly selected a region of similar length on the X chromosome and performed the same LD analysis for comparison. This region showed distinctly lower LD (Supplementary Fig. 8). In addition, LD decay analyses showed that the candidate *flatwing* region is significantly different from all other regions of the genome (Supplementary Fig. 9). These results unambiguously verify extensive and long-range linkage at the putative *flatwing* locus. Extensive genomic hitchhiking can decrease the probability that a new genetic variant is adaptive during the early stages of an adaptive bout due to effects on other traits[8,56], and the ongoing, long-range linkage disequilibrium (LD) we detected provides support for empirical observations about a range of phenotypic effects associated with *flatwing*, which are known to have both positive and negative fitness outcomes (cf. direct effects in Fig. 1a).

Intriguingly, in addition to *doublesex*, we found that several other genes exhibiting heightened expression levels in Kauai flatwing individuals were located in this highly linked genomic region and potentially implicated in various phenotypic effects linked to *flatwing* (Supplementary Table 8). For example, the linked flatwing region contained *Shaker*, a gene that regulates sex pheromone discrimination in male *Drosophila melanogaster* fruit flies[57], and *NaCP60E*, a gene involved in processing olfactory information in the same species[58] and identified as a candidate associated with divergence of cuticular hydrocarbon (CHC) profiles[36]. These observations provide a plausible genetic hypothesis for the phenomenon of delayed attraction to flatwing CHCs[59]. In addition, the *Mfsd14a* gene, which has an insect homologue identified in *Drosophila melanogaster*[60] and causes infertility in mice upon disruption[61], was identified within the linked region. Similarly, the highly conserved gene *HSD17B12*, whose haploinsufficiency leads to subfertility in female mice[62], is also located in this region and differentially expressed between flatwing and normal-wing individuals. These findings are consistent with observations from Kauai laboratory colonies of *T. oceanicus* established before 2016, in which males and females carrying the *flatwing* allele had lower sperm viability and experienced more instances of mating failure, respectively, than normal-wing individuals[44,63]. Kauai individuals carrying *flatwing* alleles show greater plasticity, manifested as more flexible and sensitive responses to social and acoustic signals[31,34]. We found several genes implicated in DNA methylation and histone modification, as well as transcriptional regulation, in the highly linked *flatwing* region, suggesting a mechanism for the observed genetic correlation between *flatwing* and enhanced plasticity. For example, *Bmi1* plays an important role in epigenetic modification and cell plasticity[64].

## Intragenomic coadaptation during a major social transition

It is well-established that transitions or other alterations to social environments can have strong phenotypic and fitness consequences, and selection arising from these transitions has been argued to be a significant evolutionary force that affects the tempo of adaptation and diversification[65,66]. Genetic responses to selection are required for the latter evolutionary processes to occur, and when a new variant under selection is the cause of social transitions, the resulting feedback process is predicted to drive intragenomic coadaptation[2,67]. There is surprisingly little empirical data that can inform this controversial

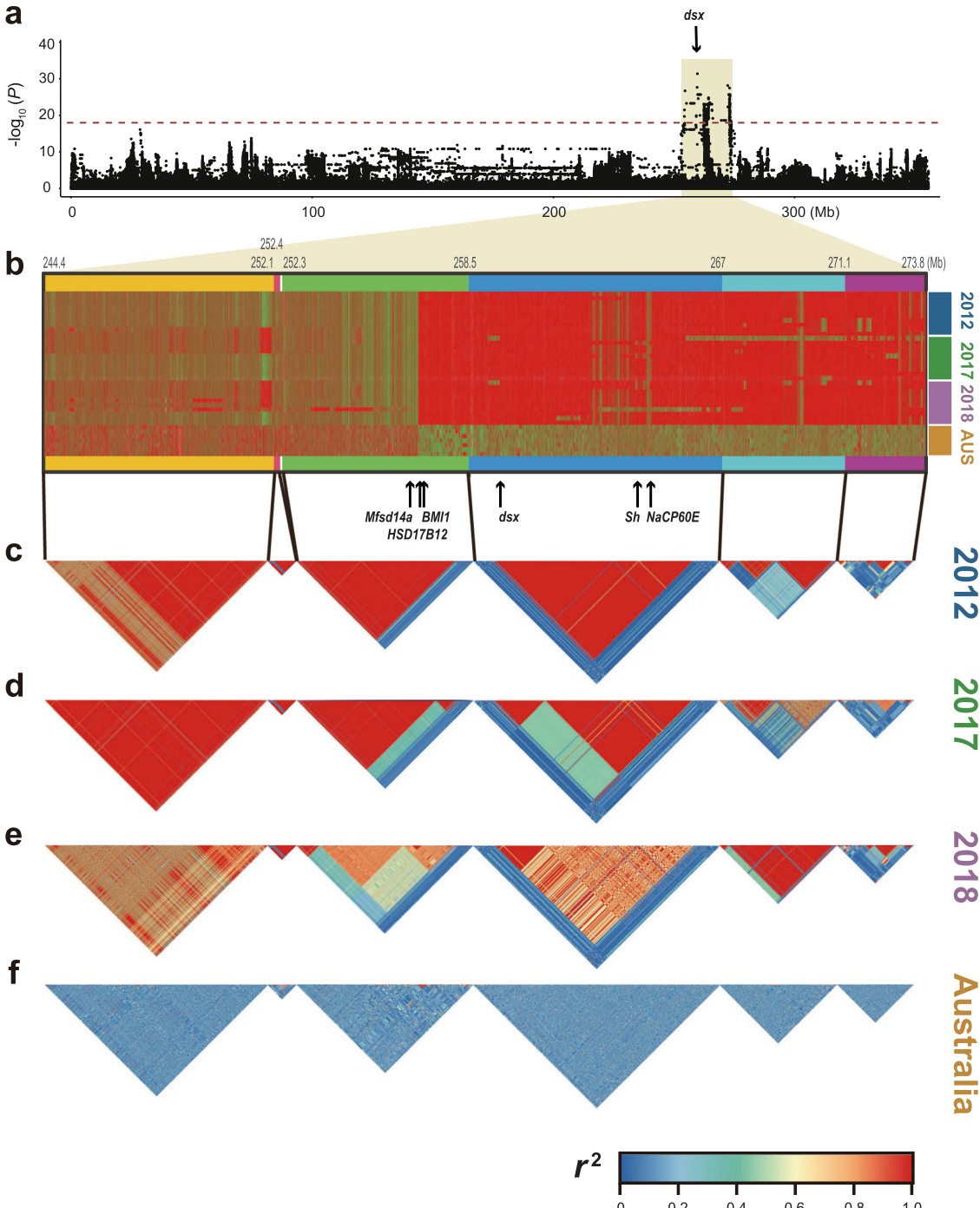

**Fig. 4 | Patterns of long-range linkage disequilibrium around the candidate *flatwing* locus at three time points of an evolutionary time-series. a** X chromosome Manhattan plot highlighting the *flatwing* locus and approximate location of *doublesex*. Shading indicates the extent of the candidate *flatwing* locus. Two-sided Fisher's Exact tests were used to calculate *P*-values. The horizontal dashed line indicates the FDR-adjusted significance threshold for multiple comparisons. Statistical details and source data are provided in Supplementary Data 4. **b** Patterns of haplotype sharing across three-time points with an Australian control group suggesting extensive long-range haplotype sharing. Rows and columns represent different individuals and SNPs respectively. SNP variants are represented as different colours. Six SNP-abundant contigs in this region and their genomic positions are shown above the panel. The positions of genes listed in the main text are indicated by black arrows. **c**–**e**. Patterns of linkage disequilibrium along the candidate *flatwing* region consistent with a hard selective sweep involving *flatwing* in the years (**c**) 2012, (**d**) 2017, (**e**) 2018. **f** Linkage disequilibrium along the same region in the Australian control group. Each contiguous block in the LD pattern corresponds to a contig to avoid any visual bias introduced by gaps in the genome assembly.

process, perhaps due to the experimental complexity of disentangling simultaneous evolutionary feedback loops (cf. Fig. 1a; see also[67]), and the specific and well-characterised set of conditions required of a study system to exclude alternative explanations. As a result of the rapid bout of adaptation in Hawaiian *T. oceanicus*, the social environment of the Kauai population changed dramatically from one dominated by a conspicuous signal conveying information about potential mating partners or rivals to one in which the mating dynamics no longer correspond in a meaningful way to social signals in the environment (Fig. 1b).

Acoustic signalling is the only method by which both male and female *T. oceanicus* locate mates from a distance. The species is nocturnal in Hawaii, which precludes mate location using long-distance visual signals[68]. Unlike some insects that use widely diffused chemical pheromones to attract the opposite sex[69], the relatively non-volatile cuticular hydrocarbons of *T. oceanicus* function in close-range sexual signalling[36,70]. Compared with other insects, singing crickets typically have a low level of aggregation, requiring them to use acoustic signals to find each other[71]. The perception of an individual cricket in the Kauai population we studied at time points 2012 and 2017 was that there were still potential mates or rivals. By 2018, the song was extinguished from the population and the acoustic perception was of total social isolation. Because the song is so central to reproductive behaviour, and alternative strategies for mate location are not readily apparent when it is lacking, we expected that the transition to a completely silent social environment imposed new selection pressure. The absence of singing males might also be expected to exacerbate fly selection on silent crickets: observed infections persist despite the absence of normal-wing targets and alternative hosts, and *flatwing* males from Kauai exhibit prolonged survival and increased spermatophore production post-infestation, distinguishing them from other populations[48].

How might these consequences exert selection to increase genomic coadaptation? In contrast with morphological studies, observing and quantifying the physiological impacts of silent social environments and changed fly infestation poses challenges. We approached the question of how such consequences might exert selection for genomic coadaptation using two methods: first, a bottom-up approach to identify candidate-selected gene sets using a combination of genome-wide selection scans, and second, a top-down approach to identify candidate genes for key selected traits over the three resequenced periods using differential gene expression data. Expression data were obtained from two published experiments. One exposed Kauai crickets to acoustic treatments mimicking selection that would be expected to drive compensatory evolutionary responses in silent populations fixed for *flatwing*[49] (Fig. 5a). The second focused on fly infestation treatments expected to be of ongoing relevance in the Kauai population[48] (Fig. 5b). In the latter dataset, samples from a non-parasitized population from the Polynesian island of Mangaia were also used to detect candidate differentially-expressed genes (DEGs) for immunological responses unique to the Kauai population[48].

Supplementary Fig. 1 provides a detailed diagram of the workflow, and an extended Supplementary Note section describes the experimental design and procedures used to obtain DEGs. All RNA-seq data had been previously published but were re-analysed here using a genome-guided approach (Supplementary Data 2, see 'Methods' for details). Following this, we performed a synthetic analysis comparing patterns of overlap between candidate DEGs identified from RNA-seq contrasts versus candidate-selected gene sets and candidate genes differentiated between years. This enabled us to test and interpret patterns of change over our sampled time intervals (Supplementary Fig. 1).

## Bottom-up genome-wide selection analyses to identify candidates under selection

We used a combinatorial enrichment approach to infer selected genomic regions accompanying the social transition from song to silence, by systematically comparing three time points across the 6 years from 2012 – 2018. Using two methods, we identified candidate genomic regions exhibiting signatures of selective sweeps, while excluding any influence of the highly linked flatwing region. First, we identified genomic regions that exhibited signatures of positive selection by comparing π log-ratios and the population differentiation $F_{ST}$ index between wild Kauai flatwing crickets and Australian normal-wing crickets that were not under selection pressure from the parasitoid fly[30,72–74]. This analysis was performed for each time point independently (Fig. 6a). The mean number of selective sweep regions at each time point was 2246, comprising 1.9 % of the cricket genome (Supplementary Table 9). Across all comparisons, a total of 1240 candidate genes under selection were identified from within the positively selected genomic regions. Of these genes, 27% were shared by all three-time points, and an additional 34% were shared between pairwise comparisons. The others were year-specific selection events (Fig. 6a). These candidate genomic regions, and sets of genes located in these regions, are referred to as "*Kauai-selected regions*" and "*Kauai-selected genes*", respectively. Gene Ontology (GO) enrichment analyses were used to summarise the biological function of this gene set (Supplementary Fig. 10). We found that genes linked to a distinct physiological response of the Kauai population to fly infestation were subject to persistent selection from 2012 to 2018. Ongoing selection on immune function due to fly infestation was corroborated by the enrichment of immune response-related GO terms in gene sets at each time point (Supplementary Fig. 10).

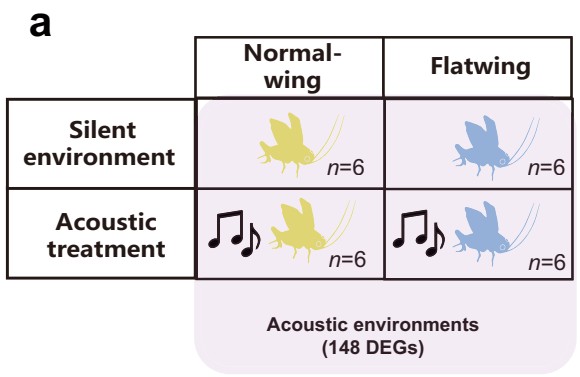

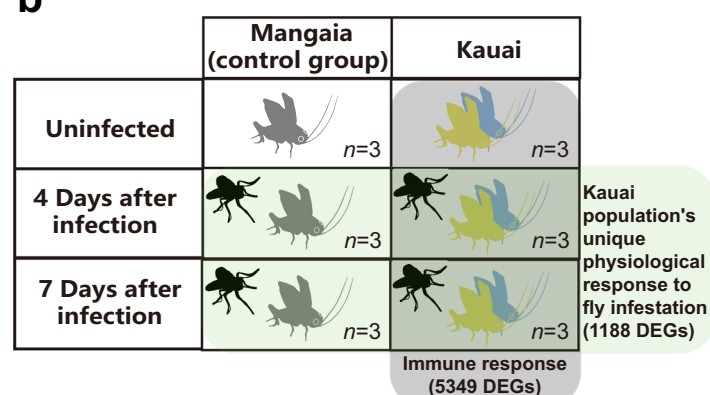

**Fig. 5 | Design of gene expression experiments that exposed crickets to conditions caused by fly selection. a** The experiment depicted on the left examined differential expression arising from a silent environment mimicking fixation of *flatwing*[49]. **b** The experiment depicted on the right tested expression consequences of infestation by *O. ochracea* parasitoid larvae[48]. The complexity of variation in social environments, and of selection that arises during development can arguably lead to unpredictable and multifarious evolutionary consequences which are difficult to observe using traditional methods. The differentially expressed genes indicated were therefore used to highlight key candidates and test whether genes related to key changes during the transition from singing to a silent population in Kauai were enriched among the genes showing signatures of selection over the same transition (See Main Text, Supplementary Fig. 1, Supplementary Note, Supplementary Data 2 and 5 for details).

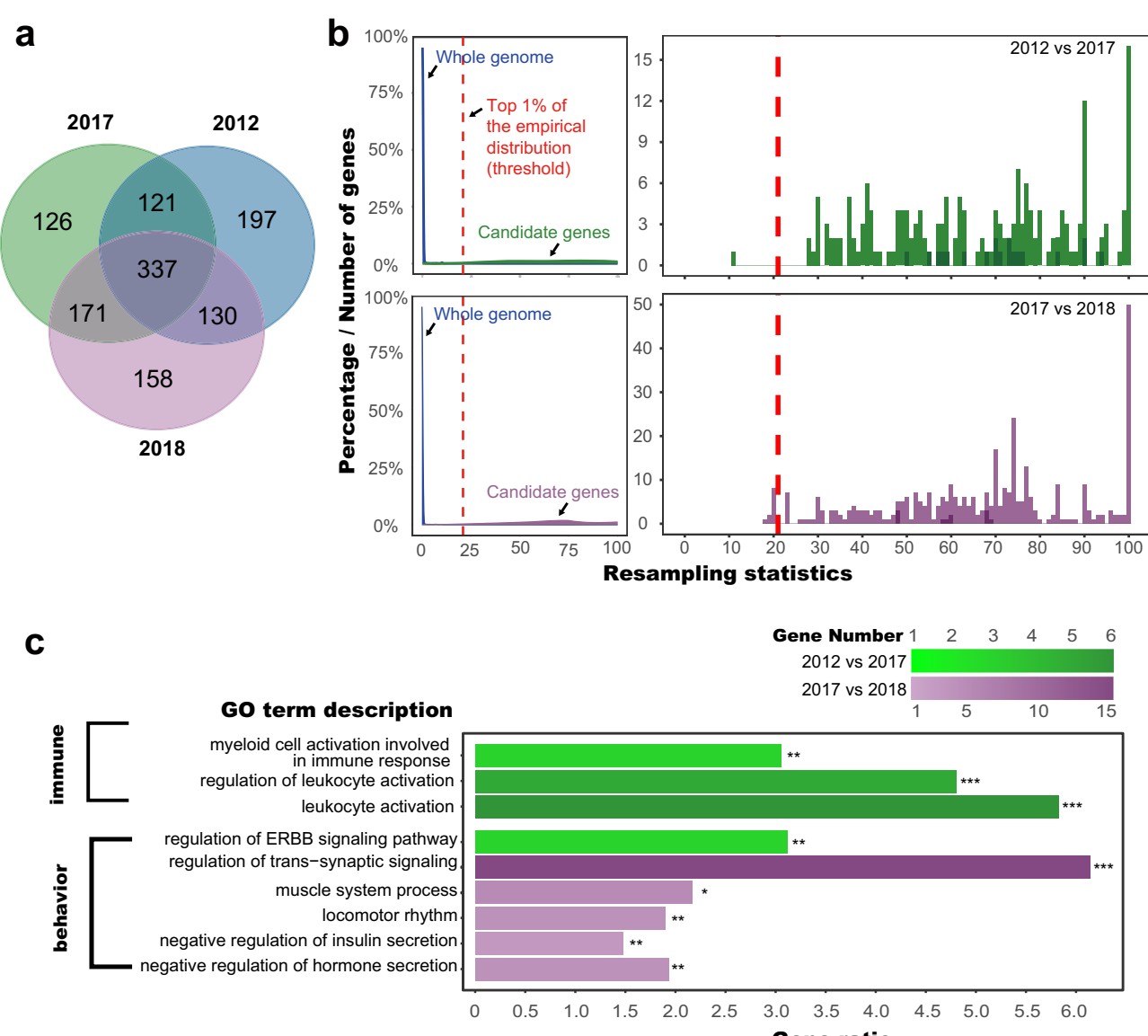

**Fig. 6 | Selection dynamics at three-time points and functional associations with altered social environments underlying genome-wide compensatory coevolution. a** Venn diagram showing shared vs. unique *Kauai-selected genes* at each time point, identified by overlapping π log-ratios and $F_{ST}$ in sliding windows (see Main Text for methodology). **b** Resampling statistics of *genes differentiated over time* (2012 vs 2017, 2017 vs 2018). Density curves (left panels) of the resampling statistic distribution of genome-wide (blue) and candidate genes (green and purple) are used to establish significance thresholds. The vertical dashed lines indicate the top 1% of empirical distributions. Interval distribution plots (right panels) focus on the resampling statistic values of only the differentiated candidate genes (top−2012 vs 2017, bottom− 2017 vs 2018). The same significance thresholds are indicated by red vertical dashed lines. **c** GO function categories with significant enrichment in the two sets of *genes differentiated over time*, with each time interval colour-coded as above (all FDR-adjusted *P* values < 0.05, two-sided hypergeometric tests, source data and statistical details are provided in Supplementary Data 8). The *x*-axis indicates the richness factor of each GO term and shading corresponds to the number of genes. Asterisks reflect the robustness of these enrichment tests after applying different filtering criteria. * significant after filtering genes with resampling statistics < top 1% ** significant after filtering genes with resampling statistics < top 0.5% *** significant after filtering genes with resampling statistics < top 0.3%. Note that GO term descriptions are described using functional terminology based on non-insect species, but all analyses are performed based on homology.

We then defined genomic *regions differentiated over time* by comparing π log-ratios and the population differentiation $F_{ST}$ index between samples collected at different time points and retaining those in the top 5% of both (see Methods). Selective sweeps are expected to reduce genetic diversity and result in enhanced differentiation[72,73]. These analyses, therefore, provided us with evidence for evolutionary transitions associated with changes in the social environment. We identified 525 genomic regions differentiated over the time interval 2012–2017, which contained 190 *genes differentiated over time* (Supplementary Table 9). Between 2017 – 2018, there were 819 genomic regions and 360 *genes differentiated over time* (Supplementary

Table 9). The statistically significant increase in the number of differentiated genes during the second interval between 2017 and 2018 (Binomial test: *P* = 0.017) could reflect the dramatic change in social environment and shifts in selection acting on mating strategy that occurred then. To exclude the possibility that stochastic variation during field sampling affected our results, we utilised an iterative resampling method to evaluate the robustness of these candidate regions and filter out less robust regions, retaining only those that passed a top 1% threshold (Fig. 6b). Functional assessment of *genes differentiated over time* based on GO annotations revealed that, during the first interval (2012 – 2017) when singing males were still present,

genes under selection were enriched for functions related to immune response and synaptic plasticity (Fig. 6c). Following the local extinction of singing males, more genes related to behavioural changes and metabolic/locomotor abilities were under selection (Fig. 6c). These enrichment tests based on gene annotations retained their significance after applying much stricter filtering criteria (Fig. 6c), indicating that such changes are robust at the population genomic level.

Multiple independent analyses confirmed that the selected regions we detected were under selection in the experimental population of interest in Kauai, and not in the Australian population used as a comparator in some analyses. First, the Australian population is not under fly selection as *O. ochracea* does not exist there, it contains no flatwing males, and it has not undergone ancient bottlenecks[30]. Second, in all of our selection analyses, all candidate regions under selection show significantly lower genetic diversity in the Kauai population and higher genetic diversity in the Australian population because we set that criterion during the analysis of nucleotide diversity (see 'Methods' section)[30,72–74]. Third, cross-population analyses for detecting selection used Australian crickets as a control group, whereas cross-timepoint analyses for detecting selection within Kauai (i.e., 2012 vs. 2017 and 2017 vs. 2018) were conducted independently from Australian samples. The functional consistency observed between the results obtained from cross-population versus cross-timepoint comparisons indicates our bottom-up genome-wide selection analyses were qualitatively reliable (Fig. 6c and Supplementary Fig. 10). Fourth, we used integrated haplotype score (iHS) and cross-population extended haplotype homozygosity (XP-EHH) statistics to further validate the robustness of selection signatures identified above using phased-genotype approaches. Of all candidate genes (*Kauai-selected genes* and *genes differentiated over time*) detected using our bottom-up approach (Supplementary Fig. 1), 74% were further validated using these phased haplotype approaches (Supplementary Table 10). In aggregate, these findings provide high confidence that the detected regions are under selection in Kauai flatwing crickets, as opposed to in Australian crickets.

## Top-down differential expression analyses to identify candidate genes for key selected traits

To identify candidate genes associated with traits likely to have experienced altered selection over the transition from singing to a silent population in Kauai, we re-analysed all published gene expression data related to two traits in *T. oceanicus*: plasticity to singing vs. silent acoustic environments, and immunological responses to parasitoid infestation (Fig. 5). In field crickets, a lack of acoustic signals in the environment has profound effects on physiological functions related to immunity, mating tactics and reproductive investment[44], as well as life-history trade-offs, locomotion and reproductive behaviour, and morphology[42,75,76]. Unlike wing morphology, which can be measured easily using standard geometric morphometric methods, the physiological effects of silent social environments and fly infestation are difficult to observe and measure directly. Gene expression patterns detected using RNA-seq provide an indirect way of quantifying such traits[49,77]. Based on these RNA-seq data, we identified a total of 5839 potential candidate genes associated with fly infection and with altered acoustic environments (Supplementary Data 5, 6). Next, we performed a synthetic analysis comparing patterns of overlap between candidate-selected gene sets with candidate DEGs to validate the functions of candidate genes identified in our bottom-up approach and evaluate their involvement in compensatory intragenomic coadaptation.

## Synthetic analysis to identify genes implicated in compensatory intragenomic coadaptation

We synthetically analysed selected genes from our bottom-up approach – i.e., *Kauai-selected genes* and *genes differentiated over time* – with differentially expressed genes detected in the top-down

approach above to test for signatures of genetic coadaptation during the transition from a singing population to a silent population. This involved testing the overlap of the bottom-up and top-down gene sets to ask whether selected candidate genes were enriched for functions relevant to compensatory adaptation during this major social transition. The existence of 20 years' worth of functional and phenotypic studies on the evolution of flatwing males enabled us to use these synthetic analyses to corroborate – or reject – a mechanistic connection between evolutionary dynamics detectable at the genome level and behavioural and physiological dynamics that have been detected at the phenotypic level.

Compared to a genetically related *T. oceanicus* population from the Cook Islands unexposed to the parasitoid fly, high-condition flatwing male crickets from Kauai survived infestation for longer and were more likely to produce a spermatophore, even after infestation, and showed significant differences in gene expression in response to infestation[48]. Fly infestation has considerable consequences for host physiological functioning, immunological responses, and gene expression[48,78,79], and consistent with this, our synthetic analyses found that DE genes associated with responses to fly infestation were significantly enriched (i.e., more likely to be located) in the *Kauai-selected gene* sets from all three-time points (Fisher exact tests: all $P < 0.05$, Supplementary Table. 11). Moreover, the number of differentiated genes associated with the Kauai population's unique physiological response to fly infestation increased over time (Supplementary Table 12, Chi-square test: $P = 0.027$), in a pattern consistent with ongoing adaptation to fly pressure. For example, the highly-conserved protein encoded by *caiap* regulates resistance to infection. It contains a highly conserved ankyrin repeat domain that was originally characterised in the *Drosophila melanogaster* protein, *Notch*, and functions in the production of signalling protein complexes called inflammasomes[80]. This gene was detected as a top candidate in all of our selection analyses contrasting Kauai with the Australian control, and it was significantly upregulated in infected cricket bodies (Fig. 7a). This gene was significantly differentiated between 2012 and 2018, consistent with continuous local adaptation as well as field observations that the parasitoid flies still exist in Kauai and silent male crickets sometimes harbour *O. ochracea* larvae[28].

In the time interval between 2012 and 2017, when singing males were still present at low frequencies (Fig. 1b), it was expected that individuals with high levels of behavioural plasticity to the social environment conferred by sensitivity to environmental stimuli would experience a selective advantage as the number of singing males was very small, and apparently decreasing. In contrast, the transition of the social environment caused by the extinction of singing males between 2017 and 2018 introduced a different, and perhaps more extreme, form of selection arising from the social environment. The perception of social isolation has profound phenotypic and fitness consequences in animals – not always negative[81], even in insect taxa not typically considered to be social, and intriguingly, often with effects on immunocompetence[82–84]. In the Hawaiian cricket population we studied, a total lack of male acoustic signals is therefore expected to be an extreme social environment which renders useless existing flexible satellite mating tactics used by silent males. Selection arising from the changed social environment at that point would presumably favour individuals that can locate mates in the absence of song, for example by increasing locomotion to moving further or more quickly in search of mates.

Studies of crickets from the Kauai population have consistently found flexibility of behaviour, physiology, and morphology in response to variation in the acoustic environment mimicking that caused by the spread of silent flatwing males, and in the aggregate, these phenotypically plastic responses appear to be unique to Kauai compared to other populations in which the abundance of flatwing vs. normal-wing crickets is less skewed. Compared with other *T. oceanicus*

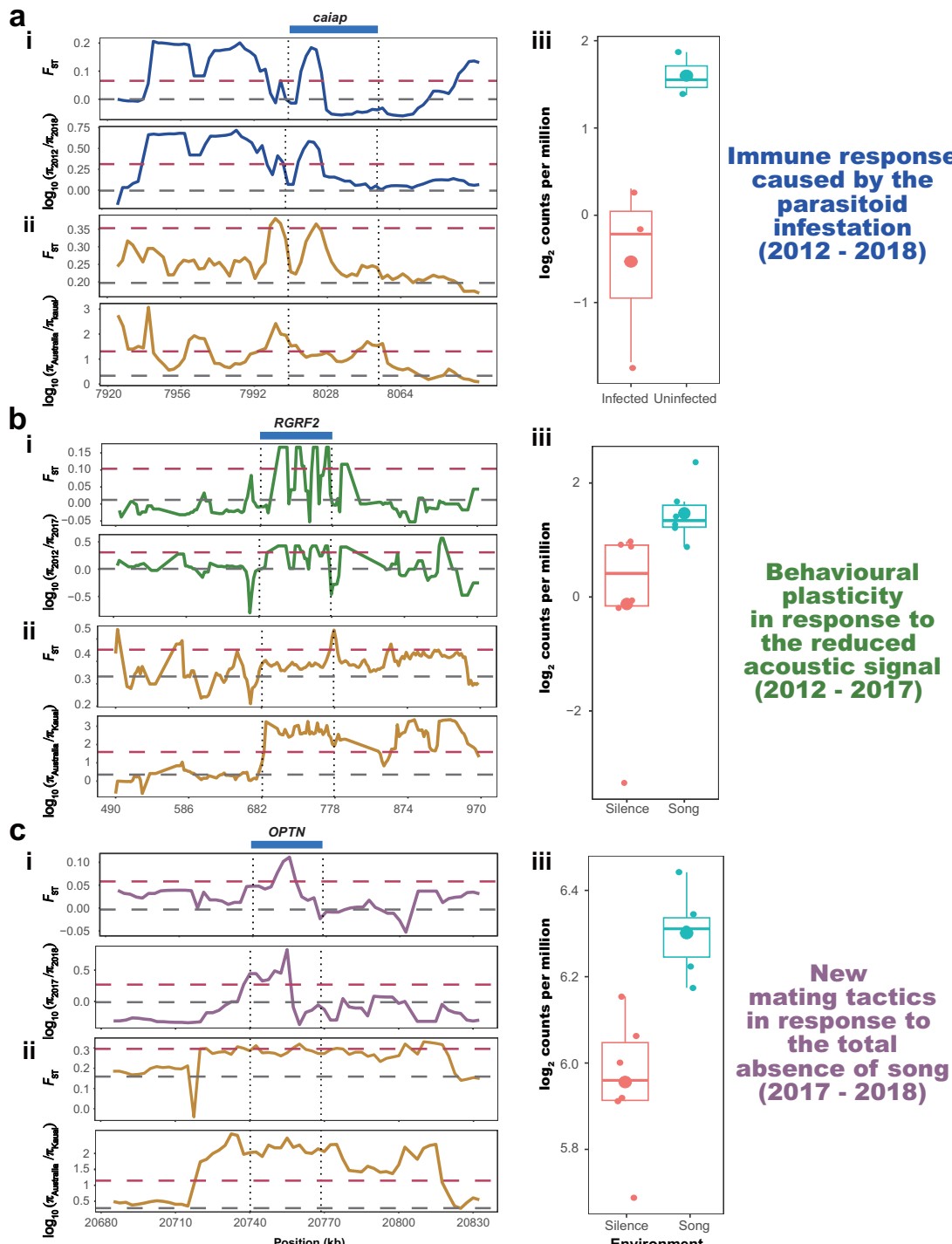

**Fig. 7 | Top candidate genomic regions and phenotypes undergoing compensatory coevolution with flatwing.** Genomic signatures of selection and expression profile for: **a** the *caiap* homologue on Scaffold 10, which is associated with response to parasitoid fly infestation, **b** the *RGRF2* homologue on Scaffold 5, which is associated with behavioural plasticity in response to reduced acoustic signals, **c** the *OPTN* homologue on Scaffold 5, associated with altered mating tactics caused by the absence of singing males. The position of candidate genes is indicated at the top of each panel, accompanied by dashed vertical lines. The flanking regions shown for each candidate gene are proportional to the length of the respective gene. $F_{ST}$ and $\pi$ were calculated using sliding-window analyses with 10 kb windows to make comparisons (**i**) between Kauai individuals collected from different years, and (**ii**)

between Kauai individuals of corresponding time points and the Australian control group. Horizontal grey dashed lines denote the mean whole-genome value for the displayed statistics in each panel, while dark red horizontal dashed lines indicate the top 5% threshold values. (**iii**) Patterns of relative expression of each candidate gene in crickets exposed to different treatments (see Main Text and Supplementary Note for details and references; for panel (**a**) $N = 3$ individuals per treatment. For panels (**b**) and (**c**) $N = 6$ individuals per treatment). Box plots show upper quartile, median, and lower quartile values, and $1.5 \times$ interquartile ranges. Large and small points show means and individual observations for each replicate respectively. Source data are provided as a Source Data file.

populations, females from Kauai are less choosy about male courtship songs[85] and show increased responsiveness to acoustic signals[31,34]. Kauai females reared in silent conditions are more responsive to male songs than if they experienced song during rearing[43], and males show increased locomotion as a response to silent social environments[35] and increased satellite behaviour[44]. Such enhanced responsiveness to acoustic signals is consistent with our evolutionary analyses based on genomic data. Among the three *Kauai-selected gene* sets (Fig. 6a), the year 2017 showed the largest number of selected genes related to environmental stimulus (Supplementary Fig. 10). In addition, the genes differentiated between 2012 and 2017 were significantly more likely than non-selected genes to be differentially expressed under different acoustic environments (Fisher exact test: $P = 0.04$). Our synthetic analyses identified several key genes related to behavioural plasticity (Supplementary Data 7). For example, *RGRF2* encodes the RasGRF2-substrate of Cyclin-dependent kinase 5, playing a vital role in synaptic plasticity, learning, and memory[86]. This gene was under continuous selection in Kauai crickets, significantly differentiated between 2012 and 2017, and differentially expressed in crickets that experienced silent vs. singing environments (Fig. 7b). These results are consistent with field observations that the year 2017 was the last year in which singing normal-wing males appear to have existed in the Kauai population, and suggests that genes with functions related to perception of acoustic signals experienced relaxed selection after 2017.

Following 2017, the perception of song became irrelevant due to the absence of singing males in the population. Laboratory tracking experiments have shown that population-level reproductive success can be compensated by increased locomotion in silent, flatwing-dominated conditions[55], leading to the prediction of greater selection pressure favouring locomotion after 2017. Consistent with this, genes differentiated between 2017 and 2018 were significantly enriched in GO categories including "muscle system process", "locomotor rhythm", "insulin secretion", and "hormone secretion" (Fig. 6c). Four candidate genes were not only significantly differentiated between 2018 and 2017 but were also differentially expressed in brain samples between different acoustic treatments (Supplementary Data 7). One of these, *OPTN*, was identified in all of our selection, differentiation, and differential expression analyses (Fig. 7c) and is essential for the development of visual perception and muscle regeneration[87]. The *D. melanogaster* orthologue *kenny* also functions in immunological defence[88]. In addition, seven *genes differentiated over time* in the interval 2017–2018, which were also differentially expressed between the Kauai population and the control population that is not under selection pressure from the parasitoid fly, are also associated with locomotor capacity (Supplementary Data 7).

## Discussion

Any model considering the role of compensatory intragenomic coevolution during adaptation is only useful if it can make falsifiable predictions. The implication is that in the absence of such coevolution, adaptation would be less likely, slower, or impossible. However, artificially arresting intragenomic coevolution is a major experimental hurdle that has yet to be overcome with the exception of work on sexual conflict in experimental populations of *Drosophila melanogaster* (e.g.[89]), and it is uncertain how many loci are expected to interact during such a process. Can evolutionary responses at one or a few key loci mitigate the direct and indirect negative fitness consequences of a genotype under selection, such as modifier loci that evolve during the spread of insecticide resistance? Or are genome-wide responses expected? To address such questions, we circumvented the experimental limitations above by taking advantage of two decades worth of field and laboratory research that has established key traits facilitating the exceptionally rapid spread and then fixation of silent male flatwing crickets in a single Hawaiian population. By minimising the possibility of confounding effects caused by differences in population genetic

parameters over successive sampling periods, we were able to characterise genomic selection in detail before, during, and after a major social transition caused by the fixation of *flatwing*. The existence of gene expression data from experimental work that exposed crickets to conditions caused by the original direct source of selection – fly infestation – and to conditions caused by the indirect source of selection arising from *flatwing's* spread – a silent social environment – enabled us to test and reject the null hypothesis that these gene sets were no more or less likely to have experienced selection during intragenomic coevolution. Instead, the gene sets were enriched for genes selected in the wild. Functional analysis of selected genomic regions lent further support to the hypothesis of genome-wide coevolution during *flatwing's* rapid spread, as opposed to other processes such as incidental environmental changes: selected genes were enriched for functions related to phenotypic responses to the social transition caused by fixation of silent crickets that were unique to the population in question. This evidence supports a process of rapid compensatory change at the genomic level, as opposed to a more general cascade of evolutionary changes such as the sequential acquisition of novel traits.

Previous work on evolved resistance to insecticides, antibiotics, and pathogens in insects, bacteria, and other microorganisms, respectively, has suggested that evolution of modifier genes ameliorates negative pleiotropy of genomic variants under selection[11,23,88,90,91], but compelling evidence remains scant, particularly in wild populations. Our findings, by contrast, implicate genome-wide coevolutionary responses to selection caused by a selected variant with profound negative consequences, not least of which is the elimination of the entire mate recognition system. The genomic breadth and apparent rapidity of this response are consistent with Pavlicev and Wagner's 'selection, pleiotropy, and compensation' model of adaptation[6]. Adaptive evolution as a direct response to novel selection thus begets further adaptive evolution to reduce indirect fitness costs of the genotype under selection, a process likely to be particularly important when adaptation is driven by de novo mutation or adaptive introgression rather than selection on standing variation. While the modern study of genomic adaptation has justifiably focused on identifying adaptation hotspots – those genes directly implicated in phenotypic responses to selection – our findings underscore the importance of the broader genomic environment in which they are expressed, both in terms of the direct genomic environment and the indirect genomic environment constituted by the local gene pool. Now may be an opportune moment to give the architects of the Modern Synthesis another hearing on one of their most controversial ideas (about which they bitterly disagreed)[1]: that adaptive evolution does not occur in a genetic vacuum, and depends as much on the compatibility of the genomic background as it does on characteristics of novel mutations themselves.

## Methods

### Cricket samples and genome sequencing

We obtained one female field cricket (*Teleogryllus oceanicus*) from a laboratory line derived from Kauai and homozygous for the *normal-wing* genotype, which had experienced multiple generations of laboratory inbreeding[36]. The study species is not subject to ethical regulations and approval of study protocols was not required. High molecular weight DNA was used to construct Oxford Nanopore Technologies (ONT) libraries and these were sequenced on the ONT PromethION platform according to the manufacturer's protocols. Following ONT base calling and read filtering, we improved single-base accuracy by polishing with previously published Illumina HiSeqX short reads from a male of the same stock population[30]. We used a third individual from the same stock population to construct Hi-C libraries following the workflow of the Arima v2 kit. Libraries were amplified and sequenced on the Illumina NovaSeq 6000 platform.

## De novo *T. oceanicus* genome assembly

Contigs were assembled from PromethION (55x) reads using wtdbg2 (v.2.5)[92]. The initial self-error correction was performed using the same ONT dataset, minimap2[93], and wtpoa-cns (19830203)[92]. BWA-MEM (v0.7.17)[94] and wtpoa-cnd[92] were then used to polish this assembly with the Illumina short read dataset (ca. 42x). These steps yielded a contig N50 of 3.05 Mb. We resolved chimeric contigs of this contig-level assembly using Hi-C paired-end reads (ca. 108x) and the 3D-DNA pipeline (v180419)[95]. Subsequently, we re-aligned the Hi-C reads to the improved contig-level assembly and filtered them to correct for erroneous mapping following the Arima-HiC Mapping Pipeline recommended by the manufacturer, with BWA (v0.7.17), SAMtools (v1.9), and Picard Tools (v2.26.2). We passed both the BAM file and the corrected version of the contig-level assembly to ALL-HiC (0.9.13) with the parameter (--RE GATC,GANTC,CTNAG,TTAA) which specified the restriction sites for enzymes used in the Hi-C experiment. ALL-HiC uses an allelic contig table generated by BLAST-based methods to remove noisy Hi-C signals[96]. We conducted additional confirmation using Purge Haplotigs (v1.1.2) with default settings to validate the allelic contig table[97]. A chromosomal-scale assembly at the scaffold level was built using the All-HiC pipeline with innovative 'prune', 'partition', 'rescue', and 'optimise' steps. We manually curated the chromosomal assembly following the Rapid Curation workflow (https://gitlab.com/wtsi-grit/rapid-curation/), which utilised Pretext-View v.0.25 for Hi-C data visualisation and interactive assembly refinement. Afterwards, gaps within scaffolds were filled using TGS-GapCloser (v 1.0.3)[98], ONT reads, and the Illumina short-read data. Three iterative polishing steps implemented by Pilon (v1.23)[99] were also conducted in the TGS-GapCloser pipeline to further improve local base accuracy.

## Sex-linked scaffold (X chromosome) confirmation

We used nf-LO (v 1.8.0), an implementation of UCSC tools and aligners such as BLAT[100], to map the polished assembly to the original PBJ-PI assembly (v1) published for this species[36] and generate the corresponding UCSC chain file, which connects the old and new coordinate systems. Linkage group 1 (LG1) was previously identified as the putative X chromosome based on marker-specific heterozygosity, sequencing coverage, and sample information obtained from genome-wide RAD-seq data at the population level[36]. After the conversion steps, Scaffold 1 was successfully anchored to LG1. We also mapped the original genome assembly (v1) to the new genome assembly (v2) using minimap2 (2.17-r941)[93] with the asm5 parameter and further validated this result. The flatwing phenotype is X-linked[28,29,101], and the *flatwing* mutation is localised to Scaffold 1 (see below). Together, these lines of evidence identify Scaffold 1 as the X chromosome.

## Repeat and gene prediction

We used RepeatModeler (v2.0.1)[102] with RECON (v1.08) and RepeatScout (v1.0.6) to construct a de novo repeat library[103]. Interspersed repeats and low complexity DNA sequences were comprehensively detected by comparing nucleotide sequences to the Dfam consensus database (v.3.1)[104] and the de novo repeat library using RMBlast (2.9.0 + )[105] and RepeatMasker (v4.1.0). We used RepeatProteinMask (v4.1.0) to identify TEs at the protein level by searching against the TE Protein Database. Unclassified repetitive elements were further classified by TEclass (v2.1.3)[106]. In addition, TRF (v4.09) was used to detect additional repeats in the assembly. These repetitive regions were masked before gene prediction[107].

SNAP (2013_11_29)[108], Glimmer-HMM (3.0.4)[109], and GENEID (v 1.4)[110] were used to generate ab initio gene sets. Gene sets from *Locusta migratoria*, *Acyrthosipon pisum* and *Drosophila melanogaster* were used as training data to obtain the best parameters for ab initio predictions[36]. The BRAKER (v 2.1.5) pipeline was also used[111]. Cleaned and trimmed RNA-seq reads generated by five previous studies (Supplementary Data 2) were aligned to the new *T. oceanicus* genome assembly (v2) using HISAT2 (v 2.2.0)[112]. SAMTOOLS (v1.9)[113] was used to sort and convert SAM files generated by HISAT2 to BAM format. These BAM files were then processed in BRAKER. Spliced alignment information was extracted from BAM files, and this transcriptome information was used to train GeneMark-ET (v4, January 2020) to generate initial gene structures. These results were then used to train AUGUSTUS (v3.3.3), integrating RNA-seq information into final gene predictions[111].

Data from 114 RNA-seq libraries were used for gene predictions, which covered four tissues, three populations, both sexes and two wing morph genotypes (Supplementary Note and Supplementary Data 2)[30,36,48–50]. We conducted the transcriptome data-based prediction using the HISAT-StringTie pipeline[114]. BAM files generated by HISAT2 and SAMTOOLS were passed to StringTie (v 2.1.2)[115], StringTie assembled these alignments into transcripts and genes for each BAM separately, and then expression levels of these genes were estimated sample by sample. These separate prediction results were merged before feeding back to StringTie to re-estimate the expression level of each gene. The re-estimated expression abundances were passed to edgeR (v3.30.3)[116] using the prepDE.py script[115]. Predicted genes that were not expressed at >1 count per million in at least three samples were filtered and removed.

Six protein sequence datasets were extracted from previously published genome assemblies and gene sets of *Locusta migratoria*[117], *Drosophila melanogaster*[118], *Gryllus bimaculatus*[119], *Laupala kohalensis*[119,120], *T. occipitalis*[121], and *T. oceanicus* (v1)[36]. These sequences were then aligned to the repeat-masked field cricket assembly (v2) using BLAST (v2.9.0)[105]. BLAST2GENE (version 17)[122], GENEWISE2 (v2-4-1)[123] and Liftoff (v1.6.3)[124], were then utilised to obtain the final homology-based gene set. Results from *ab intio*, homology, and transcriptome-based methods were integrated into a nonredundant consensus of gene structures using EVidenceModeler (EVM)[125]. We removed genes likely to be spurious, those with low EVM support, and genes only supported by a minority (≤ 2) of ab initio methods. The Programme to Assemble Spliced Alignments (PASA, v 2.4.1) was used to update this filtered list of genes and finalise the new official reference gene list for the *T. oceanicus* v.2.0 genome (i.e., the entirety of genes predicted in the whole genome)[125].

## Evaluation of genome assembly and gene sets

The completeness of the assembled genome and official reference gene list were both assessed using BUSCO v. 3.0.2 with the Arthropoda dataset[126]. To evaluate genome quality, we mapped filtered Illumina short-read WGRS data (see below) to the assembled genome using BWA-MEM (v0.7.17) with default parameters and mapped transcriptome data to this assembly using HISAT2 (v 2.2.0). Basic assembly statistics were calculated from FASTA using the assembly-stats programme (https://github.com/sanger-pathogens/assembly-stats).

## Sample preparation and whole-genome re-sequencing

The whole genomes of 47 male *T. oceanicus* field crickets were analysed in this study. Tissue samples from thirty flatwing males were collected from the same wild Kauai population in 2012, 2017, and 2018 (10 each year), in time intervals spanning the transition from a social environment with the song to a silent social environment (Fig. 1b). Ten normal-wing male Kauai field crickets from laboratory stocks derived from the same field location were also used. The latter individuals were taken from existing laboratory stocks because of the extreme scarcity of normal-wing males in the Kauai population (or indeed absence in later years). In addition, seven Australian field crickets were resequenced for comparison, because the Australian populations are not under selective pressure from the parasitoid fly. Among all 47 samples, resequencing data from 27 individuals was previously published[30]. The 20 new individuals in this study were resequenced using identical procedures.

Genomic DNA was extracted from the legs of 20 male individuals using the CTAB method. The quantity of the extracted DNA was measured using a Qubit dsDNA kit, and also using Nanodrop, 1% agarose gel electrophoresis, AATI Fragment Analyser and the Standard Sensitivity Genomic DNA Analysis Kit. DNA samples were sheared to a 450 bp mean insert size using a Covaris LE220 focused-ultrasonicator. Then, paired-end sequencing libraries were constructed, PCR amplified and sequenced on and Illumina HiSeqX following the Illumina SeqLab workflow.

### Assessment of sequence quality, read alignment, and identification of SNPs

We applied the demultiplexing, read filtering, alignment and SNP detection pipeline described in Zhang et al.[30] to process new WGRS data. Demultiplexing and adaptor trimming were carried out using bcl2fastq (v2.20, Illumina), with an allowance of 1 mismatch during the assignment of reads to barcodes. Raw reads with more than 5% unidentified nucleotides (N) or over 65% low-quality nucleotides (Phred quality value ≤ 7) were filtered[30]. All 47 *T. oceanicus* individuals were genotyped following the GATK best practice[127]. We mapped clean reads to the *T. oceanicus* reference genome using BWA (v0.7.17)[94]. The resulting files from BWA were sorted, formatted, and deduplicated using Samtools v1.10 and Picard v2.23.6. We utilised RealignerTargetCreator and IndelRealigner to enhance alignment accuracy in regions with indels[128]. HaplotypeCaller in GATK (v3.8) was used to detect raw SNPs and indels for individual samples, which were then merged using CombineGVCFs for individuals within the same population[128]. The raw SNPs of multiple populations were re-genotyped and aggregated using GenotypeGVCFs. A set of hard-filtering criteria (Qual By Depth < 2.0, Fisher Strand > 60.0, RMS Mapping Quality < 40.0, Mapping Quality Rank-Sum Test < −2.5, Read Pos Rank-Sum < −1.0) was manually selected based on density plots of the SNP dataset and implemented using SelectVariants and VariantFiltration functions[30,127,129]. Given the XX/XO (female/male) sex determination system observed in crickets, all male samples were assigned homozygous genotypes for X-linked loci[30,130]. Additionally, a Perl script was employed to filter low-quality SNPs potentially arising from sequencing errors, whereby SNPs with Phred-scaled quality scores < 30 or exhibiting abnormal depth (below one-third or exceeding three times the mean sequencing depth per individual) were filtered out[30,129,131]. For SNPs located in the X chromosome, the criteria for abnormal depth were adjusted to half those applied to autosomes. Heterozygous calls for X-linked SNPs were rectified to homozygous states in cases where one allele exhibited a depth threefold or higher than the other. In cases where neither allele was supported by a supermajority, the genotype was deemed unknown. Loci exhibiting unknown genotypes in over 25% of all individuals were excluded[30]. To mitigate potential biases caused by outgroups, SNPs were called using different subsets of individuals and subsequently utilised for subsequent analyses[30].

### Genome-wide association and linkage disequilibrium analyses

We performed a genome-wide association analysis using the new chromosome-level genome assembly. For this we used 30 flatwing individuals from Kauai and 30 normal-wing individuals originating from three different Hawaiian islands; of the latter, 10 normal-wing individuals were from the Kauai laboratory stock as described above (Supplementary Data 3)[30]. Fisher's exact tests implemented in PLINK (v1.90b6.12)[132] and Bonferroni-corrected in R were utilised to generate the standard case/control threshold for association significance and account for multiple testing[130,132]. For LD analyses, we partitioned the original SNP dataset into subsets of each time point and then converted them to haploview-recognised format using VCFtools (v0.1.16)[133]. Correlation coefficients ($r^2$) were calculated using Haploview (v 4.2) and visualised using the R function LDheatmap[134].

### Genetic structure, selection and differentiation analyses of the Kauai population

We used PLINK (v 1.90b6.12)[132] and PHYLIP (v 3.697)[135] to construct neighbour-joining phylogenetic trees, which we visualised using MEGA (v10.0.5)[136]. The smartpca programme in EIGENSOFT (v7.2.1)[137] and ADMIXTURE (v1.3.0)[138] were used to perform principal component analysis (PCA) and population structure analysis, respectively. We used VCFtools (v0.1.16)[133] to calculate genetic diversity (π), fixation indices ($F_{ST}$), and Wright's inbreeding coefficient (F).

We used a bottom-up approach to identify *candidate selected regions* (Supplementary Fig. 1). These candidate regions either underwent selective sweeps at each of the three-time points (which we term *Kauai-selected regions*), or they exhibited differentiation between different time points (which we term *regions differentiated over time*). They were identified based on a sliding-window method[30,72,73] that compared nucleotide diversities and fixation indices ($F_{ST}$) using 10 kb windows and a 2.5 kb step size. The π ratio was computed as $\pi_{Control} / \pi_K$, where $\pi_{Control}$ and $\pi_K$ represent the nucleotide diversity values for the control group and the population under selection, respectively[30,72–74]. Genomic regions containing the top 5% of $F_{ST}$ values and the top 5% of π log ratios were identified as *candidate selected regions* and any adjacent (overlapped) candidate regions were merged into a single candidate region. Then, we identified *candidate selected genes* as those that overlapped at least one candidate region. There were two types of candidate selected genes: those that underwent selective sweeps at one of the three time points in Kauai (i.e., *Kauai-selected genes*), and those that showed differentiation across timepoints (i.e., *genes differentiated over time*).

To evaluate the robustness of these candidate regions and avoid potential biases that could have been introduced through stochastic variation in field-sampled genotypes across years within the Kauai population, we performed a validation procedure using iterative resampling 100 times for each differentiation analysis. Specifically, we adopted a subsampling strategy in which 9 individuals were randomly selected without replacement from the population of 10 individuals at each time point. This step yielded a total of 10 scenarios for each time-divided group, resulting in 30 sub-sampled populations (10 scenarios * 3 time points). The differentiation analysis described in the paragraph above was then performed iteratively using these sub-sampled populations. The outcomes of these analyses were used to calculate resampling statistics for each candidate gene significantly differentiated between time points; this statistic reports the percentage of resampling iterations that support the candidate gene in question. This approach enabled us to obtain a more reliable estimate of significance for observed differentiation and selection signals. The threshold for resampling statistical significance was determined to be the top 1% of the empirical distribution of resampling statistics across the whole genome.

We used two genome-wide, phased-genotype approaches to validate the robustness of selection signatures identified from $F_{ST}$ and π log-ratio tests. First, haplotypes were reconstructed from unphased SNP genotype data using Beagle (v5.4)[139,140] with missing data imputed. Then, we computed cross-population extended haplotype homozygosity (XP-EHH)[25] and haplotype scores (iHS)[141] in Selscan (v2.0.2)[142]. Using Selscan's norm programme (v1.2.1)[143], the cricket genome was divided into non-overlapping 10 kb windows, and both XP-EHH and iHS scores were normalised with a significance threshold of 5% for detecting selection[143–145]. Positive threshold values were used for XP-EHH analysis to exclude genes under selection in Australia but not Kauai. A candidate gene was considered validated if it overlapped with a genomic region exhibiting at least one phased-genotype-based selection signature.

## RNA-seq datasets

Data from 114 RNA-seq samples were obtained from five previous studies (Supplementary Note and Supplementary Data 2), which had been deposited in the European Nucleotide Archive (ENA)[30,36,48–50]. These datasets covered four tissues: brains (tissue contained within head capsules), forewing buds of developing nymphs, thoracic sections of embryos, and general body tissue (whole bodies with heads, digestive tract, pronotum, wings, and legs removed). Three populations were represented (Kauai and Oahu in Hawaii, and Mangaia in the Cook Islands); as well as both sexes and two wing morph genotypes (*normal-wing* and *flatwing*). We used all of these sequencing data to aid gene prediction and annotation, and used 66 of the samples to establish candidate gene sets whose expression is associated with changes in complex physiological characters and validate the functions of candidate genes under selection and associated with alterations in the social environment (Fig. 5, Supplementary Fig. 1, Supplementary Note, Supplementary Data 2). Briefly, 24 brain samples from crickets subjected to acoustic environment manipulation and 18 body samples from crickets artificially infested with *O. ochracea* larvae, all sequenced on an Illumina HiSeq 2000. Brain samples were obtained from Kauai pure-breeding *flatwing* and *normal-wing* lines derived from the same laboratory stock population used for de novo genome sequencing above. As described in Pascoal et al.[49], samples were divided into two treatment groups and exposed to different acoustic environments during rearing. One group was reared in silence mimicking a population with no normal-wing males. The other group was reared under exposure to Kauai male calling song playbacks mimicking a population with a high density of singing males. These treated samples covered *flatwing* and *normal-wing* genotypes and there were 6 biological replicates each. For the 18 parasitoid-infested cricket body RNA-seq samples, a separate cricket colony originating from the same wild Kauai population and a lab-reared colony originating from the island of Mangaia were used. As described in Sikkink et al.[48], crickets were experimentally infested with *O. ochracea* larvae following previous infestation protocols. RNA samples were extracted from body tissues collected from 6 different groups (uninfected controls, four days post-infestation, and seven days post-infestation for both populations). Each group contained 3 biological replicates. The Supplementary Note and the original published articles contain additional technical details[30,36,48–50].

## Read alignment, gene expression quantification, and differentially expressed gene identification

Cleaned and trimmed RNA-seq reads were aligned to the chromosome-level *T. oceanicus* genome assembly using HISAT2 (v 2.2.1)[112] and SAMTOOLS (v 1.9)[113]. StringTie (v 2.1.4)[115] assembled these alignments based on the new official reference gene list and then estimated the expression levels of these genes for each sample. Only transcripts included in the official reference gene list were processed (with the -e parameter). The prepDE.py3 script was used to extract count information provided by StringTie and generate an edgeR-recognised matrix. Differential expression (DE) analyses were then performed at the 'gene' level in R (v 4.0.3) using edgeR (v 3.32.0)[116]. Genes with insufficient counts for downstream statistical analyses were filtered using the filterByExpr function based on count-per-million (CPM) values. Read counts were normalised by the trimmed mean of M values (TMM) implemented in the calcNormFactors function. We used multidimensional scaling (MDS) plots drawn by the plotMDS function as an unsupervised clustering method to visualise overall patterns of gene expression. To avoid bias caused by intrinsic factors varying among different RNA-seq datasets (e.g., biological lines, RNA extraction methods, sequencing platform, etc.), all differential expression analyses were performed on a study-by-study basis, and restricted to within-study comparisons. Negative binomial (NB) distributions were used to account for variability among biological replicates. Differential expression was tested using a likelihood ratio test, after which the decideTestsDGE function was used to identify differentially expressed genes (DEGs) with an FDR-corrected significance threshold of $P < 0.05$[146]. Please read Supplementary Note, Supplementary Data 5, 6 and the original published articles for more technical details.

## Functional assignment, gene content, gene ontology, KEGG enrichment, and synthetic analyses

Functional annotations of all genes predicted in the official reference gene list were assigned based on Swissport[147], TrEMBL and NR (2021/1/15) databases using a reciprocal BLAST method implemented in crbBlast (0.6.6, blast v2.5.0)[148]. Kyoto Encyclopaedia of Genes and Genomes (KEGG) terms were assigned using BlastKOALA (version 2.1) based on the KEGG (family_eukaryotes) database[149]. GO terms were assigned using GO annotations downloaded (Nov 28, 2020) from the GO Consortium[150]. Throughout this study, gene contents of candidate regions were obtained by comparing their coordinates with the new official reference gene list for *T. oceanicus* genome v.2.0. Genes overlapping the corresponding candidate regions were defined as candidate genes. Intersections of resulting candidate gene sets of selection and differentiation analyses were calculated and visualised using a Venn diagram tool (Bioinformatics & Evolutionary Genomics, Ghent University). Hypergeometric tests implemented in R (v 4.0.3) were used to perform enrichment analyses for GO terms and KEGG pathways. All reported *P* values were calculated by R unless otherwise specified.

We synthesised information from the following bottom-up and top-down approaches to identify genes implicated in compensatory genomic coevolution. A Venn diagram tool (Bioinformatics & Evolutionary Genomics, Ghent University) and Fisher's exact tests implemented in R were used, respectively, to visualise and compare candidate gene sets (i.e., *Kauai-selected genes* and *genes differentiated over time*) obtained from whole genome resequencing data with the candidate DEGs obtained from transcriptome data and test the statistical significance of these overlaps. In other words, DEG sets from the two experiments shown in Fig. 5 was separately overlapped with gene sets that had been independently identified to be under selection in Kauai, and over the temporal sampling regime, to identify loci implicated in compensatory intragenomic coadaptation (Supplementary Fig. 1). Visualisation of results was facilitated using the RectChr (BGI-Shenzhen), rgl, and ggplot2 R packages[151].

## Reporting summary

Further information on research design is available in the Nature Portfolio Reporting Summary linked to this article.

## Data availability

The genome sequencing reads, Hi-C reads, and the genome assembly for *T. oceanicus* have been deposited in the European Nucleotide Archive with the accession numbers PRJEB63577 and CAXIVR010000000. The whole-genome re-sequencing reads that support the findings of this study have been deposited in the European Nucleotide Archive with the accession numbers PRJEB63577 and PRJEB39125[30]. Previously published RNA-seq data have been deposited in the European Nucleotide Archive under accession numbers PRJEB27235[36], PRJNA636298[48], PRJNA344019[49], PRJNA283744[50], and PRJEB40088[30]. Following public databases were used in this study: Dfam consensus database [https://www.dfam.org/releases/Dfam_3.1/][104], NR database [https://www.ncbi.nlm.nih.gov/refseq/about/nonredundantproteins/], GO database [http://geneontology.org/][150], KEGG database [https://www.kegg.jp][149], Swiss-Prot and TrEMBL databases [http://www.uniprot.org/downloads][147]. Source data are provided with this paper.

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

## Acknowledgements
We thank Katherine Holmes, Tony Ly, Peter Moran, Sonia Pascoal, Michael Ritchie, John Rotenberry, Will Schneider, Suzanne Vardy, and Marlene Zuk for assistance with field sampling. Tanya Sneddon provided valuable assistance with DNA extraction. Kathryn Elmer, Carolin Kosiol, Michael Ritchie, and Jinsheng Sun provided valuable feedback on early versions of the manuscript. This study was supported with funding from the UK Natural Environment Research Council to N.W.B. (NE/T000619/1, NE/T014806/1, NE/L011255/1) and M.B. (NE/W001519/1). We are grateful for bioinformatics support from Iain Milne and the use of the UK's Crop Diversity Bioinformatics HPC, funded by the UK Biotechnology and Biological Sciences Research Council (BB/S019669/1), as well as the St Andrews Bioinformatics Unit, funded by a Wellcome Trust ISSF award (105621/Z/14/Z).

## Author contributions
Conceived the project: X.Z. and N.W.B. Designed experiments: X.Z., M.B., and N.W.B. Collected data: X.Z., J.M.D.W., A.T., S.M., J.G.R., K.L.S., S.L.B., and N.W.B. Analysed data: X.Z., J.M.D.W., A.T., S.M., P.T., J.G.R., S.Z., K.L.S., and S.L.B. Led manuscript writing: X.Z. and N.W.B.

## Competing interests
The authors declare no competing interests.
