## [Peer Review File · Nature Communications]

Temporal genomics in Hawaiian crickets reveals compensatory intragenomic coadaptation during adaptive evolutionREVIEWER COMMENTS

Reviewer #1 (Remarks to the Author):

The study by Zhang and colleagues provides an exciting, and unique, study of temporal changes in genomic variation during an episode of natural selection. The strength of the paper is in leveraging this well studied system, along with a new genome assembly and addition of genomic resequencing, to examine changes in allele frequency and differential expression across three time points. I think the paper has compelling results and would be broadly appealing to readers of Nature Communications.

However, I have a number of concerns, which involve robustness of the results. Some of these might be clarified with additional analyses or writing.

Major revisions:

Inbreeding: The authors examine a neighbor-joining tree and the nucleotide diversity, intending to address whether relatedness could be a confounding effect. Unfortunately, this is not necessarily addressing the true problem, which is that inbreeding alone might cause increased fixation across the genome, leading to false-positives in your tests of genomic selection. I think the described methods can be retained, but specific measures of inbreeding (ie FIS, or Wright's inbreeding coefficient, from F-statistics, or perhaps Tajima's D statistic measured genome-wide) should be included and used to evaluate this effect. If strong inbreeding is found, statistical thresholds for selection would need to be adjusted to account for this demographic effect. Nucleotide diversity does not necessarily capture inbreeding (ie nucleotide diversity declines more slowly than segregating sites following inbreeding).

Comparisons across time intervals: Time comparisons are made in absolute time (years), but population genetic processes operate over generational time. It is unclear how one should interpret comparisons of 2017 to 2018, for example. This comparison (? 1 generation) seems unlikely to lead to fixation or strong allele frequency differences due to selection, unless it is incredibly hard selection. I think this is especially the case with sampling variance around allele frequencies, when you have only 10 individuals.

Writing composition: There were a few places in the text where I was confused, and these could be improved for readability. Notably, I found the first paragraph of the introduction challenging. The description of "two secondary sources" (line 18) was not clearly delineated.

I would suggest revising line 23 as "As a second mechanisms, the spread of adaptive variants..." And for lines 26-28, I would clarify whether this conceptual model integrates both of the "secondary sources" or not. Its not clear as written.

Furthermore, the two "sources" are differentiated verbally, but not in terms of genetic expectations. I am not clear how they would be identified as separate mechanisms. The introduction might benefit from a sentence or two clarifying how genetic patterns might differ. Finally, in the third paragraph of the introduction, I found lines 56-58, and lines 62-63, to be contradictory at first. "2003 to near -but no complete- fixation within roughly 20 generations." I assume this is 2003-2023, with annual generations, but perhaps it is not. A few simple changes in the text could clarify that you are describing a temporal sequence of selection, with eventual fixation occurring in 2017.

Lines 778: It remains unclear how you filtered for allele depth and missingness for all the data. In general, an important question I had was about the reliability of genotype calls, given the depth of sequencing. GATK often performs best with very high coverage sequencing, or by implementing VQSR with additional data. Stating the parameters is a bit vague, so additional sentences to clarify their function in SNP filtering can be helpful to the reader.

Minor revisions:

Line 37: change to "systems of experimental evolution"

Line 209: When a species names starts a sentence, the genus needs to be spelled out (even if the full Latin name appears earlier in the text).

Line 300: sharing of haplotypes should be replaced with the word "linkage" I think.

Line 301: how long is LD in terms of KB?

Line 303: I did not follow the relevance of this sentence invoking Fisher's geometric model. It was distracting. Perhaps you can rework this sentence.

Line 362: fix as 'these transitions have' instead of "these has"

Line 372: "demographics" is a bit generic, when you might instead use "mating dynamics"

Line 384: remove "dramatic", hyperbole and subjective

Line 407: replace 'the distinct' with 'a distinct'
Line 482: insert "to" before "validate the functions"
Lines 493-496: This was redundant, I would remove it.
Line 496: At this point, your findings are not yet made clear to the reader.
Line 503: "profound", how so? Perhaps revise this broad claim.
Line 516: should be "adaptation"
Line 596, and thereafter: With stating the year, it would make most sense to order them chronologically, so "2017 and 2018" here.
Line 605: rephrase as "locomotory capacity"
Line 620: edit as "confounding effects caused by"
Line 627: I think you want to rephrase this "selected and differentiated". Selection implies significantly differentiated, so the terms are redundant. I might revise as "selected and differentially expressed" to capture the results of your study.
Lines 751-752: rephrase as "resequenced for comparison, because the Australian populations are not under selective pressure.."

Reviewer #2 (Remarks to the Author):

Zhang et al have carried out an interesting study on Hawaiian crickets that have become silent over the period of only few years due to selection pressures from a parasitoid fly. They combine previously published RNAseq and resequencing data with a new chromosome-level reference genome and new resequencing data of silent crickets from two time points (2012+2018). They frame the study in terms of compensatory mutations, a topic with so far very limited empirical evidence. It has already been shown previously, that silent males are parasitised less but are also less effective in attracting females due to loss of long-range signalling (signing) and reduced short-range signalling (pheromones). Therefore, compensatory mutations that increase the mating success rates are expected (increased locomotion and satellite mating tactics in populations where some males still sing). The authors show that genes showing differential expression between flatwing and normal-winged crickets also show signatures of selection between shifts of the social environment of the crickets from acoustic communication to silence.

The novelty of this study compared to the many previous studies on this topic should be highlighted more clearly. Yes, the authors have a new reference genome. However, in their previous Nature Communications paper (Zhang et al. 2021, reference 30), they already show some of the same data mapped to a chromosome-level reference genome. The genomic region underlying the flatwing phenotype was already known previously and the gene expression data is reused from previous studies. It seems to me that the key point the authors are making is that the genes differentially expressed between flatwing and normal wing crickets are under selection in the genome and thus indicate compensatory responses across many genes. However, it is currently not straightforward to extract this information from the manuscript. Many of the analyses are also not described with enough detail or have some strong short-falls that are not discussed. The manuscript requires major rewriting and additional analyses but has the potential to be very impactful.

Regarding the gene expression analyses:

The reporting of the differential gene expression results is difficult to follow. There is a lot of information presented that was already covered in the Introduction but the information about what is compared for differential gene expression is too limited. Without having the information on what is compared exactly, it is also difficult to understand the implications. For instance: «seven genes under selection between the years of 2018 and 2017 and differentially expressed between populations (Kauai vs Mangaia) are also associated with locomotion capability (Supplementary Data 6)». What is Mangaia? Are the crickets winged there or not? How many genes were differentially expressed in total between 2018 and 2017? The years refer to Kauai individuals I assume? Is 7 genes associated with locomotion more than expected? Supplementary Data 6 does not provide answers. In fact, I cannot even find the 7 genes in Supplementary Data 6. In the Discussion on L 623, the authors write: "gene expression data from experimental work that exposed crickets to conditions caused by the original direct source of

selection". That sounds great. Reading the results, it is not clear that such data was used and I cannot see the results on this. Throughout the Results/Discussion section it remains unclear what the experimental design for the RNA studies was, what data sets were compared in this study for differential expression analyses and how the different analyses were combined to find genes of interest. The data is described a bit in the Methods, but it remains unclear in the Methods what comparisons were done and how they were combined to identify candidate genes.

Regarding the selection statistics:

The flatwing individuals of this study are compared to Australian samples for LD, FST and pi ratios. However, one of the problems of this approach is that genes under selection in the Australian population would strongly influence these statistics. Given that the authors use resequencing data of 30 wildtype males in the GWAS, I think it would be good to rerun the same statistics with this additional «control» or with their Oahu wildtype samples that the authors have previously sequenced (Zhang et al. 2021). I would like to see if the same regions show up as being under selection and thus that the selection is indeed on the flatwing individuals and not the Australian ones. For the pattern of increased LD, the problem is even greater. LD is not only affected by selection but also by population structure. If only individuals with the same haplotype are used (flatwing), I am actually surprised that LD should be particularly high. The authors do not currently show if this region is any different from the rest of the genome. The authors compare LD in the flatwing region of the Hawaiian flatwing crickets to the Australian crickets. However, the Australian crickets might just have lower LD in general, everywhere in the genome. I would like to see a statistic like XP-EHH comparing the silent crickets with the wildtype crickets from the GWAS analysis. If a sweep happened in the doublesex region, we would expect that this region is an outlier in XP-EHH and that it shows longer haplotypes in the silent crickets than in the wildtype one. In addition to the FST statistics, I would also want to see statistics e.g. based on SFS (selscan) or on long-range haplotypes (e.g. iHS) that are performed in each population separately.

Some of the slightly unclear phrases and terms in the abstract are not found in the manuscript, e.g. "combinatorial enrichment" or "adaptation begets adaptation". If kept in the abstract, these should be used again in the main text and elaborated on.

Why did the flatwing males arise only recently? Is the parasitoid fly new or is the flatwing mutation new?

L. 70 Citation for the presence of flies in all-silent populations?

L. 149: unclear what they authors mean here with "genomic architecture".

L. 218: Is this true outside the flatwing gene? This gene should be removed for the PCA as it might drive the genomic structuring observed.

L. 210: Given how similar the study 30 is to this one, it would be good to quickly summarise their findings and state what the authors have analysed newly here.

Fig. 3c: This figure is very hard to read. Better plot 2D PCA or at least facilitate the position of the dots with vertical lines connecting dots to the bottom plane. Also the squares for normal winged ones and circles for flatwing males should be used here to make it easier to read.

Fig. 3b: There seems to be a rooting problem here. Use mid-point rooting to avoid that the Australian individuals form a polytomy with the Hawaiian clade.

Fig. 3b: This figure is not colour-blind safe. Please change the colourscheme. Australia and 2017 cannot be distinguished by colourblinds, nor 2012 and 2018. Labelling the major clades would also help to make this figure clearer.

L306: LD is higher in regions of low recombination without selection. I would want to see a XP-EHH.

L. 299: It is very difficult to judge if this is relevant. It is possible that LD is generally higher in

Hawaiian crickets than in Australian ones. Does this region stand out compared to the rest of the genome? Are recombination rates decreased here?

Fig. 4: The flatwing region on the X chromosome looks very much like there are structural variants involved. This should be discussed. Alternatively, the peaks might be due to the multiple genes the authors suspect to be under selection. However, the zoom-in in Fig 4b does not allow us to see if the genes in that region mentioned in the text (Shaker, NaCP60E, etc) are found on these peaks. Also in that Figure, I would want to see LD across the entire region to see if the three peaks are in LD as expected in the case of structural variants. For this LD, I would want to see LD across the 30 wildtype and the 30 flatwing individuals used in the GWAS analysis. I am actually surprised that one would see strong LD if only flatwing individuals are used given how recent the sweep is and that it looks like a single haplotype in Fig. 4b.

L. 354: How is it decided which allele is derived and which one is ancestral?

L. 401: The sharing of the same regions could also be due to selection in the Australian crickets. The authors should use additional populations to compare the Kauai flatwing crickets against to rule out this alternative.

L. 396: Where regions where π was decreased in the Australian crickets also used? These would imply selection in the Australian crickets.

L. 420: it is unclear to me how bootstrapping could remove sampling bias. What kind of sampling bias are the authors referring to here?

Fig5b: "Precentage" should be "Percentage"

Fig. 5c: gene ratio not ratio

Fig. 5d: It is very unclear to me what the message is here. Is this showing that the two tissue types differ (L487)? But how could I see the tissues here? What does D4, D7, Fw, Nw etc mean? Abbreviations should be explained.

Fig. 4b: It seems as if the x axis in the left panels was unlabelled. The label bootstrap value should be moved to the middle.

L. 490: What does "synthetically analysed" mean here? This needs more explanation.

L. 514-515: turn the parts of the sentences around. First saying what you found and then what you not found.

L. 506: What does "an increasing number of genes" mean?

Fig. 6a: It is very difficult to judge if these F_{ST} peaks are indeed higher than expected. Perhaps the horizontal lines show the mean or is it a threshold of 5% top quantile? They need explanation.

Fig. 6: How were these regions chosen?

Fig. 6ii) Should the F_{ST} here really be between all Hawaiian crickets and the Australian ones? It seems that using only the individuals of the later time point of the comparison should be compared to the Australian crickets as we would expect selection to only start after the transition (e.g. not to be present in 2012).

L. 556: this information is repetitive as it was already shown in the Introduction and in the previous section

L. 603: "seven genes under selection" is a very strong formulation. I would call them candidates for selection. Just a simple F_{ST} difference does not necessarily mean that they really are under selection as it could also be an inversion and some drift or a low-recombination region

experiencing strong background selection.

L. 604: What does a difference between Kauai vs Mangaia imply? What is Mangaia? Once one has read the methods and reread the results, it becomes clear that these are the Cook island populations that are mentioned on L499 but one should not have to do such detective work to get this key information. In L499 it does not become evident that a Cook Island population was used for the differential gene expression.

L. 646: citation for the disagreement?

L. 687: citation for the original assembly? What is the quality of this genome?

L. 717: Should be supplementary data 2, not 1. The abbreviations are impossible to understand.

L. 716: It is not clear here that this data was published previously. Please provide citations.

L. 754: Make it clear here which samples are new.

L. 774-448: No sequencing depth filter? Are homozygous sites missing? If yes, how could pi be calculated???

L. 777: What does "fixed the heterozygous calls of X-linked SNPs" mean?

REVIEWER COMMENTS

Reviewer #1 (Remarks to the Author):

The study by Zhang and colleagues provides an exciting, and unique, study of temporal changes in genomic variation during an episode of natural selection. The strength of the paper is in leveraging this well studied system, along with a new genome assembly and addition of genomic resequencing, to examine changes in allele frequency and differential expression across three time points. I think the paper has compelling results and would be broadly appealing to readers of Nature Communications.

Thank you for your positive assessment and enthusiasm for our study.

However, I have a number of concerns, which involve robustness of the results. Some of these might be clarified with additional analyses or writing.

Major revisions:

Inbreeding: The authors examine a neighbor-joining tree and the nucleotide diversity, intending to address whether relatedness could be a confounding effect. Unfortunately, this is not necessarily addressing the true problem, which is that inbreeding alone might cause increased fixation across the genome, leading to false-positives in your tests of genomic selection. I think the described methods can be retained, but specific measures of inbreeding (ie FIS, or Wright's inbreeding coefficient, from F-statistics, or perhaps Tajima's D statistic measured genome-wide) should be included and used to evaluate this effect. If strong inbreeding is found, statistical thresholds for selection would need to be adjusted to account for this demographic effect. Nucleotide diversity does not necessarily capture inbreeding (ie nucleotide diversity declines more slowly than segregating sites following inbreeding).

We thank the reviewer for raising this issue. We have calculated Wright's inbreeding coefficient from F -statistics to test whether inbreeding exists. The inbreeding coefficient F of the wild-caught flatwing Kauai samples, inbred laboratory Kauai samples, and the wild-caught Australian control samples were compared (Supplementary Fig. 6). We found that the wild flatwing Kauai individuals ($F = -0.011$) and the wild Australian individuals ($F = 0.008$) showed similar F values and the F values for wild flatwing Kauai samples across each sampling year remain stable (ANOVA, $P = 0.593$, Supplementary Table 6). In contrast, the inbred laboratory line of normal-wing individuals ($F = 0.134$) showed significantly higher F values than these wild flatwing individuals (Fisher's Least Significant Difference test, $P = 0.0291$, Supplementary Fig. 6), serving as an effective 'positive control' for inbreeding.

These results suggest that there is no inbreeding occurring in the wild Kauai population, nor differences in inbreeding across sampling years that could confound our analyses. This is also consistent with our previous observations indicating significant gene flow shared between the Kauai population and other natural populations ¹.

To further confirm this conclusion and exclude the possibility that the Australian control group is experiencing inbreeding, we utilized unpublished whole-genome resequencing data from a wild *T. emma* population (the sister species of Hawaiian field crickets) and found similar results: No difference was observed between the inbreeding coefficient of the wild *T. emma* population and the wild flatwing Kauai *T. oceanicus* population used in this study (ANOVA, $P = 0.080$).

Supplementary Figure 6. Comparisons of Wright's inbreeding coefficient for three time points in wild-caught Kauai flatwing crickets, plus two control groups. a. The inbreeding coefficient for each sampling year remains stable within the Kauai population. **b.** Data from the wild outbred Australian population and the inbred normal-wing laboratory stock originally derived from Kauai were used for comparison. The inbreeding laboratory line showed significantly higher F values than wild flatwing individuals. Horizontal lines indicate medians, boxes interquartile ranges, and whiskers the data range. Asterisk indicates significance at $P < 0.05$; see Main Text for details. Additional statistical details are provided in Supplementary Table 7.

In summary, the Kauai wild population is not experiencing inbreeding that would warrant adjustment of our statistical thresholds for selection (Lines 189-197, 691-692).

References

- 1 Zhang, X., Rayner, J. G., Blaxter, M. & Bailey, N. W. Rapid parallel adaptation despite gene flow in silent crickets. *Nat Commun* **12**, 50 (2021).

Comparisons across time intervals: Time comparisons are made in absolute time (years), but population genetic processes operate over generational time. It is unclear how one should interpret comparisons of 2017 to 2018, for example. This comparison (? 1 generation) seems unlikely to lead to fixation or strong allele frequency differences due to selection, unless it is incredibly hard selection. I think this is especially the case with sampling variance around allele frequencies, when you have only 10 individuals.

Thank you very much for bringing this important omission to our attention. The generation time for this species is approximately four generations per year. We now make this explicit in our Introduction (lines 64-66), noting that the comparisons between 2017 and 2018 span approximately four generations which is ample time for changes in allele frequencies due to selection.

“The crickets breed continuously with approximately four generations per year, indicating a rapid sweep to near – but not complete – fixation in fewer than 20 generations.”

Writing composition: There were a few places in the text where I was confused, and these could be improved for readability. Notably, I found the first paragraph of the introduction challenging. The description of “two secondary sources” (line 18) was not clearly delineated.

I would suggest revising line 23 as “As a second mechanisms, the spread of adaptive variants...”

Thank you for the suggestion. We have edited this paragraph from line 19 to make the language more direct.

And for lines 26-28, I would clarify whether this conceptual model integrates both of the “secondary sources” or not. Its not clear as written.

It integrates both – clarified at lines 26 - 28.

Furthermore, the two “sources” are differentiated verbally, but not in terms of genetic expectations. I am not clear how they would be identified as separate mechanisms. The introduction might benefit from a sentence or two clarifying how genetic patterns might differ.

This is an interesting question. We are not certain whether they are different in terms of genetic expectations – is ‘direct’ pleiotropy causing altered phenotypes in individuals carrying an adaptive variant different from more ‘indirect’ pleiotropy, where effects of the adaptive variant are exerted via social environments? We have clarified the language at line 27 to avoid giving the impression that we differentiate such genetic expectations in the present study.

Finally, in the third paragraph of the introduction, I found lines 56-58, and lines 62-63, to be contradictory at first. “2003 to near -but no complete- fixation within roughly 20 generations.” I assume this is 2003-2023, with annual generations, but perhaps it is not. A few simple changes in the text could clarify that you are describing a temporal sequence of selection, with eventual fixation occurring in 2017.

We re-wrote this section with a new paragraph starting at line 63, verbally spelling out the timeline in Figure 1b as it is central to understanding the study. We regret omitting the crickets’ generation time in our original manuscript – now noted at line 65 – as it is an important detail.

Lines 778: It remains unclear how you filtered for allele depth and missingness for all the data. In general, an important question I had was about the reliability of genotype calls, given the depth of sequencing. GATK often performs best with very high coverage sequencing, or by implementing VQSR with additional data. Stating the parameters is a bit vague, so additional sentences to clarify their function in SNP filtering can be helpful to the reader.

We have edited our manuscript to clarify these important details (lines 664-674). We used a well-established protocol following the GATK best practices – the same protocol used in our previous work¹. This includes steps for filtering allele depth and missingness. We are sorry that we omitted these details in our original manuscript. In this revised version, all details have been elaborated:

Given the XX/XO (female/male) sex determination system observed in crickets, all male samples were assigned homozygous genotypes for X-linked loci^{1,2}. Additionally, a Perl script was employed to filter low-quality SNPs potentially arising from sequencing errors, whereby SNPs with Phred-scaled quality scores < 30 or exhibiting abnormal depth (below one-third or exceeding three times the mean sequencing depth per individual) were filtered out^{1,3,4}. For SNPs located in the X chromosome, criteria for abnormal depth were adjusted to half those applied to autosomes. Heterozygous calls for X-linked SNPs were rectified to homozygous states in cases where one allele exhibited a depth threefold or higher than the other. In cases where neither allele was supported by the supermajority, the genotype was deemed unknown. Loci exhibiting unknown genotypes in over 25% of all individuals were excluded¹. To mitigate potential biases caused by outgroups, SNPs were called using different subsets of individuals and subsequently utilized for subsequent analyses¹.

References

- 1 Zhang, X., Rayner, J. G., Blaxter, M. & Bailey, N. W. Rapid parallel adaptation despite gene flow in silent crickets. *Nat Commun* 12, 50 (2021).
- 2 Kim, K. W. *et al.* Genetics and evidence for balancing selection of a sex-linked colour polymorphism in a songbird. *Nat Commun* 10, 1852 (2019).
- 3 Qiu, Q. *et al.* Yak whole-genome resequencing reveals domestication signatures and prehistoric population expansions. *Nat Commun* 6 (2015).
- 4 Wang, Z. *et al.* Chromosome-level genome assembly and population genomics of *Robinia pseudoacacia* reveal the genetic basis for its wide cultivation. *Commun Biol* 6, 797 (2023).

Minor revisions:

Line 37: change to “systems of experimental evolution”

Changed as suggested (line 39).

Line 209: When a species names starts a sentence, the genus needs to be spelled out (even if the full Latin name appears earlier in the text).

Changed as suggested (line 160).

Line 300: sharing of haplotypes should be replaced with the word “linkage” I think.

Changed as suggested (line 241).

Line 301: how long is LD in terms of KB?

We now note that the entire region in high LD spanned approximately 20 Mb and included several breakpoints (line 243).

Line 303: I did not follow the relevance of this sentence invoking Fisher's geometric model. It was distracting. Perhaps you can rework this sentence.

Rewritten for clarity (lines 250-251).

Line 362: fix as 'these transitions have' instead of "these has"

Changed as suggested (line 280).

Line 372: "demographics" is a bit generic, when you might instead use "mating dynamics"

Changed as suggested (line 290).

Line 384: remove "dramatic", hyperbole and subjective

Changed as suggested (line 303).

Line 407: replace 'the distinct' with 'a distinct'

Changed as suggested (line 341).

Line 482: insert "to" before "validate the functions"

Changed as suggested (line 397).

Lines 493-496: This was redundant, I would remove it.

We reduced and combined this with the next sentence – see next comment (lines 414-415).

Line 496: At this point, your findings are not yet made clear to the reader.

We have clarified this is a goal of the study (lines 414-415).

Line 503: "profound", how so? Perhaps revise this broad claim.

Rephrased (lines 421 - 422).

Line 516: should be “adaptation”

Corrected (line 434).

Line 596, and thereafter: With stating the year, it would make most sense to order them chronologically, so “2017 and 2018” here.

Changed as suggested (line 477).

Line 605: rephrase as “locomotory capacity”

We have changed this to “locomotor capacity” (line 486).

Line 620: edit as “confounding effects caused by”

Changed as suggested (line 502).

Line 627: I think you want to rephrase this “selected and differentiated”. Selection implies significantly differentiated, so the terms are redundant. I might revise as “selected and differentially expressed” to capture the results of your study.

Rewritten for clarity (line 509).

Lines 751-752: rephrase as “resequenced for comparison, because the Australian populations are not under selective pressure..”

Changed as suggested (line 639).

Reviewer #2 (Remarks to the Author):

Zhang et al have carried out an interesting study on Hawaiian crickets that have become silent over the period of only few years due to selection pressures from a parasitoid fly. They combine previously published RNAseq and resequencing data with a new chromosome-level reference genome and new resequencing data of silent crickets from two time points (2012+2018). They frame the study in terms of compensatory mutations, a topic with so far very limited empirical evidence. It has already been shown previously, that silent males are parasitised less but are also less effective in attracting females due to loss of long-range signalling (signing) and reduced short-range signalling (pheromones). Therefore, compensatory mutations that increase the mating success rates are expected (increased locomotion and satellite mating tactics in populations where some males still sing). The authors show that genes showing differential expression between flatwing and normal-winged crickets also show signatures of selection between shifts of the social environment of the crickets from acoustic communication to silence.

The novelty of this study compared to the many previous studies on this topic should be highlighted more clearly. Yes, the authors have a new reference genome. However, in their previous Nature Communications paper (Zhang et al. 2021, reference 30), they already show some of the same data mapped to a chromosome-level reference genome. The genomic region underlying the flatwing phenotype was already known previously and the gene expression data is reused from previous studies.

Thank you for these comments. While containing many of the same types of sequencing data, and taking advantage of several previously-published datasets in addition to a large volume of new data, the objectives of these two studies are different.

In Zhang et al. 2021¹, we did not use, nor claim to have used, a chromosome-level reference genome. Although our original work identified the X chromosome and autosomes by constructing a high-density linkage map, only 35.6% of the genome was anchored to this map. Additionally, the scaffold N50 of the old reference genome was 62.6 kb. In our new study, over 98% of the genome was anchored to 14 pseudochromosomes with a scaffold N50 of 137 Mb and contig N50 was 3.05 Mb. We have added a sentence to our revised manuscript stating, *this new chromosome-level genome assembly is a considerable improvement for this species*² (Supplementary Fig. 2) [lines 137-138], to emphasize the novelty of our study in terms of genome assembly.

The genomic region underlying the flatwing phenotype was already known previously, but as identifying this was not an objective of the current study, this knowledge does not undermine the novelty of our work. We have edited the MS in several places to avoid giving readers this impression (e.g. lines 102-108, 230, 234-235, 321).

However, in addition to wing venation, *flatwing* also causes dramatic regulatory and phenotypic disruptions such as feminization of male pheromones². The causes of these disruptions remain unknown. It is unclear whether these effects are caused via pleiotropy, genomic hitchhiking, or genetic coadaptation during a major social transition. These causes could not be explored in our previous study due to the limited continuity of v.1 of the cricket reference genome².

The primary aim of this study was to test for compensatory intragenomic coadaptation during adaptive evolution. To achieve this goal, the genomic signatures caused directly by the *flatwing* genotype (i.e., pleiotropic or genetically coupled effects) needed to be detected first and then excluded from our main analyses. With this aim in mind, we re-performed genome-wide association analyses using a larger sample of resequenced wild crickets, investigated the genomic consequences of *flatwing*'s rapid adaptive spread (i.e., super-long LD), and most importantly, excluded the genomic regions affected by *dsx*

directly from the following analyses. Consequently, only the candidate regions caused by compensatory intragenomic coadaptation were considered in the subsequent analysis.

We thank the reviewer for raising this issue, as this distinction is critical to the interpretation of our study. We now incorporate the text in the above paragraph at lines 201-209.

References

- 1 Zhang, X., Rayner, J. G., Blaxter, M. & Bailey, N. W. Rapid parallel adaptation despite gene flow in silent crickets. *Nat Commun* 12, 50 (2021).
- 2 Pascoal, S. *et al.* Field cricket genome reveals the footprint of recent, abrupt adaptation in the wild. *Evol Lett* 4, 19-33 (2020).

It seems to me that the key point the authors are making is that the genes differentially expressed between flatwing and normal wing crickets are under selection in the genome and thus indicate compensatory responses across many genes. However, it is currently not straightforward to extract this information from the manuscript. Many of the analyses are also not described with enough detail or have some strong short-falls that are not discussed. The manuscript requires major rewriting and additional analyses but has the potential to be very impactful.

We are sorry for the confusion. We appreciate your comments and suggestions, which have highlighted confusing elements of our original writing, particularly for readers who might have a vague impression of our previous work on this system and become distracted by wanting to know the causative genetic difference between flatwing and normal-wing crickets.

Here is how we have clarified these issues:

1. Our goal is to demonstrate how 'adaptation begets adaptation'. This is an important concept summarising the more technical description of 'compensatory intragenomic coevolution'. We have now defined it clearly in the abstract at lines 13 - 15:

“Our results demonstrate how ‘adaptation begets adaptation’: changes to the sociogenetic environment accompanying rapid trait evolution can generate selection provoking further, compensatory adaptation.”

2. To accomplish this goal, we explored temporal patterns of genomic selection on gene sets relevant to such compensatory responses. These analyses focused on the population after flatwing crickets had invaded and spread. The temporal sampling spanned the transition from when flatwing males comprised ca. 95% of the population to 100% fixation. This is important, because we were interested in identifying genomic selection associated with the conspicuous shift in the social environment (silence) brought about by fixation of flatwing. In so doing, our analyses aimed to reveal compensatory intragenomic coadaptation that arises during adaptive evolution. We clarify this logic in a rewritten section starting at line 97.

3. One key point that we must stress is that, unlike a previous study of ours ¹, this new study does not focus on further refining our detection of causative genetic differences between flatwing and normal-wing individuals in Kauai. We do double-check that our original association patterns still hold with the new genome assembly, which they do (line 230), but further resolution is not obtainable at this time. This is an interesting question we are actively working on, but it is not the topic of this study. We have made clarifying edits to ensure readers do not interpret this as a goal (lines 203 - 209).

4. To address the concern the reviewer raised about clarifying the analytical workflow, we have included a new flow diagram in the supplementary document to explain the design and highlight the goals of our study (Supplementary Figure 1).

Supplementary Figure 1. Schematic workflow for this study summarizing the core goal, resource development, data sources, methodological procedures, and interrelations. *Goal:* The overall scientific goal of the study is indicated by the black star. *Resources:* Purple rectangle indicates annotated reference genome resource developed in this study. *Data:* Squares indicate sequence data utilized in this study. Those shaded in blue denote whole-genome resequencing (WGRS) data from field sampling in Kauai, and icons outlined in grey represent previously published data. Green shading represents differentially expressed genes obtained in this study from previously-published reads (see Fig. 5d and Supplementary Note for details). *Procedures:* Orange ovals indicate important intermediate steps and contrasts. Arrows show connections between data, experimental procedures, and conclusions.

A more detailed summary of our key points is as follows:

- (1) By generating a chromosome-level genome assembly and mapping whole-genome resequencing data to it, we demonstrate that there is no distinct difference among the three time points at the whole-genome level. Thus, we are able to treat all three time points as the same population from a genetic point of view in our analyses, providing confidence that

patterns we detect in subsequent analyses reflect changes in selection as opposed to simple changes in population demography through migration or inbreeding. This enables us to detect highly resolved patterns of adaptation at the genomic level in the next section, entitled "*Temporal population genomic dynamics of rapidly evolving wild crickets*" in our manuscript.

(2) By exploring the effects caused by long-range linkage disequilibrium around the putative flatwing locus, we can exclude them from subsequent analyses. This is an important point, because our goal is to detect intragenomic coevolution that compensates for the indirect negative effects of *flatwing*'s fixation; we do not want this assessment to be confounded with hitchhiking genes. We now make this unambiguous by motivating these steps in our study at the beginning of the section reporting "*Genomic consequences of flatwing's rapid spread*", starting at line 201:

"Because the flatwing genotype invaded and spread in Kauai so recently, genes in its flanking regions are prone to long-range genomic hitchhiking and may be the cause of negative indirect fitness consequences of its rapid spread^{1,2}. Our goal was to detect intragenomic coevolution that compensates for indirect negative effects of flatwing's fixation, both as a direct consequence to individuals carrying flatwing, and as an indirect consequence of its changes to the social environment (Fig. 1a). Therefore, we first assessed the extent of hitchhiking in regions flanking the candidate flatwing locus and related gene function in these regions to known phenotypic associations with flatwing. Second, we excluded this region from subsequent downstream analyses of intragenomic coadaptation arising from changes to the sociogenetic environment brought about by flatwing's fixation to avoid confounding the two (Supplementary Figure 1).

It is a by-product that the results of this exploration coincidentally explained several phenomena that have been previously observed in the wild. This aspect is mainly demonstrated in the "*Genomic consequences of flatwing's rapid adaptive spread*" section.

(3) Based on the findings of the previous two points, we then evaluate "*Genetic coadaptation during a major social transition.*" The results from part 1 enable us to perform cross-timepoint comparisons, indicating that the candidate regions under selection we detected were caused by year-specific effects instead of sampling bias. The results from part 2 distinguish the genomic regions/fitness effects caused by the invasion of the *flatwing* variant itself from genetically encoded secondary adaptations. Building on this foundation, in part 3, we combine the genomic signature of selection with gene expression data obtained from two experiments (i.e., acoustic treatment and infection treatment, also see Figure 5d in our revised manuscript) to test whether selected regions are enriched for genes likely to be involved in compensatory adaptation to sociogenetic brought about by *flatwing*'s fixation.

Importantly, the third key point we aim to demonstrate in our manuscript is not related to "*genes differentially expressed between flatwing and normal wing crickets*" as proposed by the reviewer. These genes are only utilized to annotate the reference genome and highlight key candidates associated with *flatwing* and its hitchhiking regions. In addition to the new Supplementary Fig. 1, we rewritten the last paragraph of our introduction section to assist readers in understanding the design of this study (starting at line 95).

References

- 1 Zhang, X., Rayner, J. G., Blaxter, M. & Bailey, N. W. Rapid parallel adaptation despite gene flow in silent crickets. *Nat Commun* 12, 50 (2021).
- 2 Pascoal, S. et al. Field cricket genome reveals the footprint of recent, abrupt adaptation in the wild. *Evol Lett* 4, 19-33 (2020).

Regarding the gene expression analyses:

The reporting of the differential gene expression results is difficult to follow. There is a lot of information presented that was already covered in the Introduction but the information about what is compared for differential gene expression is too limited. Without having the information on what is compared exactly, it is also difficult to understand the implications. For instance: «seven genes under selection between the years of 2018 and 2017 and differentially expressed between populations (Kauai vs Mangaia) are also associated with locomotion capability (Supplementary Data 6)”. What is Mangaia? Are the crickets winged there or not? How many genes were differentially expressed in total between 2018 and 2017? The years refer to Kauai individuals I assume? Is 7 genes associated with locomotion more than expected? Supplementary Data 6 does not provide answers. In fact, I cannot even find the 7 genes in Supplementary Data 6.

Thank you for this constructive criticism. We acknowledge that the RNA-seq data used in this study were complex. Below we describe how we clarified the overall experimental design for readers. This has introduced some slight redundancy throughout the manuscript, but we think it is helpful as readers benefit from recurrent ‘signposting’ about what is being described and why.

Although all of the previous RNAseq studies focused on the Kauai flatwing population, different control groups were used in each. Mangaia is an island of the Cook Islands, near Hawaii, where all crickets are normal-winged as there is no parasitoid fly present, similar to Australia. We now include a thoroughly-detailed description of all of these datasets in a Supplementary Note section, directing readers to it in an expanded section of the Main Text describing the rationale for the RNAseq-based candidate gene sets (lines 395-406). We think it is important that readers do not have to refer to the Supplement to gain the key information you request, so we also generated a new figure (Fig. 5d) to better describe how the RNAseq analyses were designed. This replaces the previous Multidimensional scaling (MDS) plot of gene expression data, which, as alluded to in other comments, was of limited value. Additionally, more detailed results of all differential expression analyses are now provided in Supplementary Data 2, 5, 6; for example, details of candidate genes described in the Main Text can be found in Supplementary Table 6 by searching their Gene ID or gene name (UniProt Entry Name).

Regarding the question "Are the 7 genes associated with locomotion more than expected?", this observation was obtained through manual inspection of the functional annotations of key candidate genes rather than through enrichment analysis. However, we are confident that it is biologically meaningful description of genes upon which selection is acting after the fixation of flatwing crickets and ensuing silence across the population. Our results revealed that during the second time interval (2017 – 2018), genes under selection related to behavioural changes and metabolic/locomotor abilities were significantly enriched (Fig. 5c). Additionally, enrichment analysis based on overlapping the “DEGs related to Kauai population’s unique physiological response” with the “candidate genes differentiated

over time” (note that these latter candidate genes refer to the ones located in genetic regions differentiated over time, rather than DEGs) showed that genes associated with the Kauai population's unique physiological response to fly infestation were differentiated over time (Chi-square test $P = 0.027$) (line 425).

In the Discussion on L 623, the authors write: “gene expression data from experimental work that exposed crickets to conditions caused by the original direct source of selection”. That sounds great. Reading the results, it is not clear that such data was used and I cannot see the results on this. Throughout the Results/Discussion section it remains unclear what the experimental design for the RNA studies was, what data sets were compared in this study for differential expression analyses and how the different analyses were combined to find genes of interest. The data is described a bit in the Methods, but it remains unclear in the Methods what comparisons were done and how they were combined to identify candidate genes.

Your questions have helped us realize that the complexity of the datasets used in our study required further explanation. See above the responses that describe our extensive clarifying revisions. To summarise, they include:

- (1) A new figure (Fig. 5d) to summarize the experimental design for the RNA studies used in the "Genetic coadaptation during a major social transition" part of our manuscript. This new figure, combined with the new flow diagram which explains the design and highlights the goals of our study (Supplementary Fig. 1), addresses your questions regarding "what the experimental design for the RNA studies was, what datasets were compared in this study for differential expression analyses, and how the different analyses were combined to find genes of interest."

Figure 5d: Design of gene expression experiments that exposed crickets to conditions caused by fly selection. The experiment depicted on the left examined differential expression arising from a silent environment mimicking fixation of flatwing¹. That on the right tested expression consequences of infestation by *O. ochracea* parasitoid larvae². The complexity of variation in social environments, and of selection that arises during development can arguably lead to unpredictable and multifarious evolutionary consequences which are difficult to observe using traditional methods. The differentially expressed genes indicated were therefore used to highlight key candidates and test whether genes related to key changes during the transition from a singing to a silent population in Kauai were enriched among the genes showing signatures of selection over the same transition (See Main Text, Supplementary Figure 1, and Supplementary Note).

- (2) We have added a comprehensive supplementary text section to describe the most important details of previous RNA-seq experiments (Supplementary Note). Although we believe these details are sufficient for readers to understand our manuscript thoroughly, we have extensively signposted readers to the original articles with citations should they wish to access more technical details (e.g. lines: 748, 770, 1378).

- (3) According to the author guidelines for Nature Communications, the Methods section cannot appear before the Discussion section. We have added the note “see methods” and “Supplementary Note” in some sentences of the Results section to help readers find the details of these experiments more easily (e.g. lines: 405, 408, 1331, 1378).

References

- 1 Pascoal, S. *et al.* Increased socially mediated plasticity in gene expression accompanies rapid adaptive evolution. *Ecol Lett* **21**, 546-556 (2018).
- 2 Sikkink, K. L., Bailey, N. W., Zuk, M. & Balenger, S. L. Immunogenetic and tolerance strategies against a novel parasitoid of wild field crickets. *Ecol Evol* **10**, 13312-13326 (2020).

Regarding the selection statistics:

The flatwing individuals of this study are compared to Australian samples for LD, FST and pi ratios. However, one of the problems of this approach is that genes under selection in the Australian population would strongly influence these statistics.

Thank you very much for raising this. We are very confident that the selection is on the flatwing Kauai individuals, not the Australian ones. Our conclusion is supported by four main reasons:

- (1) **The eavesdropping parasitoid fly does not exist in Australia. Hence, selection caused by the flies cannot exist there. Similarly, there are no silent crickets in the Australian population. So, the selection caused by the social transition to a silent environment cannot exist. These two facts make the Australian population a perfect control group for our selection analysis.**
- (2) **In our original manuscript, we combined both π ratios and F_{ST} values to detect candidate regions under selection. Although significant F_{ST} values cannot indicate which population is under selection, significant π ratios can provide us with such information. Specifically, a selective sweep often reduces genetic diversity (π), which can be used as an indicator of genetic regions under selection^{2,3,4}. All of our selection analyses were performed specifically to identify only those candidate regions that showed significantly lower genetic diversity in the flatwing Kauai population (i.e., the experimental group) and higher genetic diversity in the Australian population (i.e., the control group), and we now make this clearer in the Methods section at line 697. These results are inconsistent with a scenario in which the Australian population was under selection.**
- (3) **We performed the requested iHS and XP-EHH statistics to verify our analysis, and 74% of our original candidate genes were further supported by at least one of these methods (Supplementary Table 10), indicating relatively high robustness of our results. Similar to the π ratio test, XP-EHH statistics can also inform which population is under selection by providing negative and positive values. For that reason, we only used positive threshold values in our analysis, indicating that the Kauai population is under selection, not the Australian one.**

Supplementary Table 10 | Validation of putative genes under selection using phased-genotype-based approaches (iHS and XP-EHH; see Main Text for details).

	2012	2017	2018	2012 vs 2017	2017 vs 2018	Summary
Candidate gene under selection with phased genotypes	561	597	606	289	145	2198
Candidate genes supported by additional selection statistics	430	455	463	176	93	1617
Verification rate (%)	77	76	76	61	64	74

- (4) **The conclusions of our selection analyses were drawn from multiple independent sources of evidence, precisely to avoid relying solely on cross-population analyses with the Australian control group (i.e. Kauai vs. Australia). For example, cross-timepoint comparisons *within* Kauai (i.e., 2012 vs 2017, 2017 vs 2018) were also conducted independent of data from Australian samples. The functional consistency observed between results from cross-population versus cross-timepoint comparisons strongly supported our conclusion that “changes to the sociogenetic environment accompanying rapid trait evolution can generate selection provoking further, compensatory adaptation”.**

We briefly spell out the rationale at lines 185-189 and in a new paragraph at line 366:

*“Genetic diversity in all years was significantly lower in the Kauai population than the Australian population (two-sided t-tests, all $P < 2.2 \times 10^{-16}$), consistent with the fact that the Australian population is not under fly selection as *O. ochracea* does not exist there, contains no flatwing males, and has not undergone ancient bottlenecks¹.“*

*“Multiple independent analyses confirmed that the selected regions we detected were under selection in the experimental population of interest in Kauai, and not in the Australian population used as a comparator in some analyses. First, the Australian population is not under fly selection as *O. ochracea* does not exist there, it contains no flatwing males, and it has not undergone ancient bottlenecks¹. Second, in all of our selection analyses, all candidate regions under selection show significantly lower genetic diversity in the Kauai population and higher genetic diversity in the Australian population because we set that criterion during the analysis of nucleotide diversity (see ‘Methods’ section)¹⁻⁴. Third, cross-population analyses for detecting selection used Australian crickets as a control group, whereas cross-timepoint analyses for detecting selection within Kauai (i.e. 2012 vs. 2017 and 2017 vs. 2018) were conducted independent from Australian samples. The functional consistency observed between the results obtained from cross-population versus cross-timepoint comparisons indicate our bottom-up genome-wide selection analyses were qualitatively reliable (Figure 5c, Supplementary Fig. 10). Fourth, we used integrated haplotype score (iHS) and cross-population extended haplotype homozygosity (XP-EHH) statistics to further validate the robustness of selection signatures identified above using phased-genotype approaches. Of the original candidate genes, 74% were validated using these phased haplotype approaches (Supplementary Table 10). In aggregate, these findings provide high confidence that the detected regions are under selection in Kauai flatwing crickets, as opposed to in Australian crickets.”*

References

- 1 Zhang, X., Rayner, J. G., Blaxter, M. & Bailey, N. W. Rapid parallel adaptation despite gene flow in silent crickets. *Nat Commun* 12, 50 (2021).
- 2 Liu, Z. et al. Genomic mechanisms of physiological and morphological adaptations of *Limestone langurs* to Karst habitats. *Mol Biol Evol* (2019). 930.
- 3 Clarkson, C. S. et al. Adaptive introgression between *Anopheles* sibling species eliminates a major genomic island but not reproductive isolation. *Nat Commun* 5, 4248 (2014). 932.
- 4 Yang, J. et al. Whole-genome sequencing of native sheep provides insights into rapid adaptations to extreme environments. *Mol Biol Evol* 33, 2576-2592 (2016).

Given that the authors use resequencing data of 30 wildtype males in the GWAS, I think it would be good to rerun the same statistics with this additional «control» or with their Oahu wildtype samples that the authors have previously sequenced (Zhang et al. 2021). I would like to see if the same regions show up as being under selection and thus that the selection is indeed on the flatwing individuals and not the Australian ones.

We appreciate your suggestion of using Oahu individuals as another control group. However, this is not feasible. Oahu cannot be used as a control group because this population is also facing selective pressure caused by parasitoid flies. In addition, the Oahu population is facing a different social environment, in which singing males and flatwing males coexist in the wild. Our previous paper detected a pattern of balancing selection in the Oahu population¹. All these factors make these individuals inapposite for use as a control group. The normal wing individuals used in our GWAS also cannot be used in the selection analysis due to their laboratory origins (line 679-680). With the additional analyses provided above, we are satisfied that the selection we detect is on flatwing individuals in Kauai. We think it would be interesting to compare temporal dynamics across multiple flatwing populations to assess whether compensatory adaptation occurs convergently or independently, but unfortunately we do not have the data to do this.

References

- 1 Zhang, X., Rayner, J. G., Blaxter, M. & Bailey, N. W. Rapid parallel adaptation despite gene flow in silent crickets. *Nat Commun* 12, 50 (2021).

For the pattern of increased LD, the problem is even greater. LD is not only affected by selection but also by population structure. If only individuals with the same haplotype are used (flatwing), I am actually surprised that LD should be particularly high. The authors do not currently show if this region is any different from the rest of the genome. The authors compare LD in the flatwing region of the Hawaiian flatwing crickets to the Australian crickets. However, the Australian crickets might just have lower LD in general, everywhere in the genome.

This is a valid point and one we were interested in solving. To do so, we randomly selected a region with the same length on the X chromosome and performed the same LD analysis for comparison (Supplementary Fig. 8). What if this was a quirk of the particular random region of the X that we chose? To exclude this, we generated an LD decay plot for the Kauai population at each timepoint. The

resulting plots show that the LD pattern of the candidate flatwing region containing *dsx* is significantly different from all other regions of the genome (Supplementary Fig. 9).

Supplementary Figure 8. Linkage disequilibrium along a randomly chosen region of the X chromosome of similar length to the linkage block containing *flatwing* reported in the Main Text. The same three time points of the evolutionary time-series are compared. Each contiguous block in the LD pattern corresponds to a contig to avoid any visual bias introduced by gaps in the genome assembly.

Supplementary Figure 9. Decay patterns of linkage disequilibrium flanking the candidate *flatwing* region containing the *dsx* locus (red) compared to genomic background (blue) illustrated by mean pairwise r^2 values for each sampled timepoint of the wild Kauai population.

Both analyses unambiguously support our original findings, which we now explain at line 244- 249.

“To exclude the possibility that this pattern arose due to generally high LD genome-wide, we randomly selected a region of similar length on the X chromosome and performed the same LD analysis for comparison. This region showed distinctly lower LD (Supplementary Fig. 8). In addition, LD decay analyses showed that the candidate flatwing region is significantly different from all other regions of the genome (Supplementary Fig. 9). These results unambiguously verify extensive and long-range linkage at the putative flatwing locus.”

I would like to see a statistic like XP-EHH comparing the silent crickets with the wildtype crickets from the GWAS analysis. If a sweep happened in the *doublesex* region, we would expect that this region is an outlier in XP-EHH and that it shows longer haplotypes in the silent crickets than in the wildtype one. In addition to the F_{ST} statistics, I would also want to see statistics e.g. based on SFS (selscan) or on long-range haplotypes (e.g. iHS) that are performed in each population separately.

Thank you for your suggestion. The wild-type crickets from the GWAS analysis cannot be used as a control group for selection analysis. Due to the rarity of normal-wing males in the Kauai population, it was impossible for us to collect enough normal-wing samples in Kauai. Therefore, we included laboratory specimens from stock populations derived from previous collections at the same location; this enabled us to breed enough of the rare morphs for re-sequencing. The GWAS, therefore, was performed between 30 wild Kauai flatwing males and 30 normal-wing males from Kauai lab stock (inbreeding lines), wild Hilo, and wild Oahu populations as clarified at lines 678-680.

Although it is inappropriate for us to use these individuals as a control group to perform selection analysis (see above), we have used the Australian population as a control group and used the software selscan to re-perform the selection analysis following your suggestions. Our results show that the *dsx* region contains significant XP-EHH outliers, thereby confirming your expectation and further supporting the original conclusion of our previous paper that used F_{ST} and π ratio statistics ¹.

Supplementary Figure 7. Distribution of standardized cross-population extended haplotype homozygosity (XP-EHH) scores in the highly-linked, flatwing-associated region of the X chromosome containing *doublesex*. The black horizontal dashed line indicates the mean background value for XP-EDD scores, while the red horizontal dashed line indicates the top 5% threshold of XP-EHH scores. The scores passing the top 5% are plotted in purple.

We now briefly report the result of this new analysis at line 234 - 235. However, we would like to emphasize that the main goal of our manuscript is to test patterns of genomic selection after flatwing individuals had already dominated the wild cricket population (Supplementary Figure 1) – not fine-map the causative *flatwing* locus or examine genomic patterns of selection on it. Re-confirming the

region of the *flatwing* locus in the new chromosome-level reference genome enabled us to double-check earlier findings, but was primarily useful because we wanted to exclude this region and the phenotypic effects caused by genomic hitchhiking from the subsequent analyses of compensatory intragenomic coadaptation (see Supplementary Figure 1). We have opted to refrain from further discussion in the MS regarding the selection signature of this highly-linked region and its formation, as it is not within the scope of this study and does not impact the results or conclusions drawn.

References

1 Zhang, X., Rayner, J. G., Blaxter, M. & Bailey, N. W. Rapid parallel adaptation despite gene flow in silent crickets. *Nat Commun* 12, 50 (2021).

Some of the slightly unclear phrases and terms in the abstract are not found in the manuscript, e.g. "combinatorial enrichment" or "adaptation begets adaptation". If kept in the abstract, these should be used again in the main text and elaborated on.

We modified line 325 to flag “combinatorial enrichment” to readers; the procedure is described in detail in the paragraphs that follow. “Adaptation begets adaptation” is now defined in the Abstract at line 14 and referred to and discussed at line 524.

Why did the flatwing males arise only recently? Is the parasitoid fly new or is the flatwing mutation new?

We clarified at line 54 that the parasitoid fly is known to have coexisted with crickets at this location prior to the emergence of the flatwing morph. We ultimately do not know why flatwing males arose so recently.

L. 70 Citation for the presence of flies in all-silent populations?

Done (line 79)

L. 149: unclear what they authors mean here with "genomic architecture".

Rewritten for clarity (line 116-118).

L. 218: Is this true outside the flatwing gene? This gene should be removed for the PCA as it might drive the genomic structuring observed.

Yes, all PCA and tree analyses have been done at the autosomal level, which means that the flatwing locus and its flanking regions have been removed from the PCA. Now noted at in the figure legend at line 1232 and line 1235.

L. 210: Given how similar the study 30 is to this one, it would be good to quickly summarise their findings and state what the authors have analysed newly here.

These studies are completely different. We now note at line 162 that ref. 30 examined parallel evolution of flatwing crickets across the Hawaiian archipelago.

Fig. 3c: This figure is very hard to read. Better plot 2D PCA or at least facilitate sing the position of the dots with vertical lines connecting dots to the bottom plane. Also the squares for normal winged ones and circles for flatwing males should be used here to make it easier to read.

Thank you for your suggestions. We have re-generated Fig. 3c by adding vertical lines connecting dots to the bottom plane. Two panels of the 2D PCA have also been plotted (Supplementary Fig. 4). All of these figures did not separate wild individuals collected from different years, which is consistent with our original description.

Fig. 3c. Three-dimensional PCA scatter diagram showing genetic relationships among wild Kauai individuals based on autosomal SNPs derived from WGRS data. Vertical lines link dots to the base to aid visualisation. See Supplementary Figure 4 for a 2-dimensional version.

Supplementary Figure 4. Two-dimensional principal component (PC) plots showing genomic distance of wild Kauai samples across years. **a.** PC 1 (x-axis) vs. PC 2 (y-axis) **b.** PC 1 (x-axis) vs. PC 3 (y-axis). Blue = Kauai individuals sampled in 2012, green = Kauai individuals sampled in 2017, purple = Kauai individuals sampled in 2018.

Fig. 3b: There seems to be a rooting problem here. Use mid-point rooting to avoid that the Australian individuals form a polytomy with the Hawaiian clade.

Thank you for catching this, we have re-generated this tree (Fig. 3b).

Fig. 3b: This figure is not colour-blind save. Please change the colourscheme. Australia and 2017 cannot be distinguished by colourblinds, nor 2012 and 2018. Labelling the major clades would also help to make this figure clearer.

Thank you for catching this, we have re-generated this figure (Fig. 3).

At the very beginning of this project, we altered all figures to use the orange, blue, green and purple colours for the Australia, and Kauai 2012, 2017, and 2018 groups following a colour-blind friendly guideline. However, as you noticed in our original Fig. 3b we neglected to adjust brightness of each colour or change of the colour of the normal-wing group. We have now altered the latter in all figures using a light blue with cross-hatching pattern, fixed the brightness of the samples depicted in the tree, labelled the major clades as suggested, and tested the figure using an online colour-blindness simulator: <https://pilestone.com/pages/color-blindness-simulator-1>

L306: LD is higher in regions of low recombination without selection. I would want to see a XP-EHH. These which are known to have both positive and negative fitness outcomes (cf. 308 direct effects in Fig. 1a)

Please see our responses above regarding XP-EHH and revisions starting at line 234 which cover this point.

L. 299: It is very difficult to judge if this is relevant. It is possible that LD is generally higher in Hawaiian crickets than in Australian ones. Does this region stand out compared to the rest of the genome? Are recombination rates decreased here?

Please see our detailed response above and text at lines 244 - 249 which covers this issue about LD. To recap, we randomly selected a region with the length on the X chromosome and performed the same LD analysis. This comparison (Supplementary Figure 8, 9) indicates a dramatic difference. We also compared LD decay in this region vs. the rest of the X and find it stands out very prominently. Furthermore, in Kauai these regions stand out compared to the rest of the genome whereas in the Australian control, there are no differences between this region and other regions (Supplementary Figure 8, 9, see above). These lines of evidence definitively support our original findings.

Fig. 4: The flatwing region on the X chromosome looks very much like there are structural variants involved. This should be discussed. Alternatively, the peaks might be due to the multiple genes the authors suspect to be under selection. However, the zoom-in in Fig 4b does not allow us to see if the genes in that region mentioned in the text (*Shaker*, *NaCP60E*, etc) are found on these peaks. Also in that Figure, I would want to see LD across the entire region to see if the three peaks are in LD as expected in the case of structural variants. For this LD, I would want to see LD across the 30 wildtype and the 30 flatwing individuals used in the GWAS analysis. I am actually surprised that one would see strong LD if only flatwing individuals are used given how recent the sweep is and that it looks like a single haplotype in Fig. 4b.

We are grateful for the interest in these questions, which we are actively investigating in ongoing research. However, the aim of this study is not to determine the involvement of SVs in flatwing or to fine-map it as we have neither the data nor the scope to accomplish this in the current manuscript. To avoid inadvertently misleading readers, we have added clarifying text in a new paragraph at line 201.

Regarding "the zoom-in in Fig 4b does not allow us to see if the genes in that region mentioned in the text (*Shaker*, *NaCP60E*, etc.) are found on these peaks": The precise location of these genes has been provided in Supplementary Table 8, and in our revision, we have noted the location of genes in revised Fig. 4 following your suggestion.

Regarding "I would want to see LD across the 30 wildtype and the 30 flatwing individuals used in the GWAS analysis": LD across the 30 flatwing individuals used in the GWAS analysis is shown in Figure 4c. As described in our responses above, however, normal-wing individuals cannot be used as a control group in the LD or selection analyses because they are either from a different population subject to different patterns of selection (e.g. Oahu) or are affected by laboratory inbreeding (e.g. lines 192, 267, 539).

L. 354: How is it decided which allele is derived and which ones is ancestral?

Our original wording was unclear. On line 1273 we re-wrote the figure legend to:

"SNP variants are represented as different colours."

L. 401: The sharing of the same regions could also be due to selection in the Australian crickets. The authors should use additional populations to compare the Kauai flatwing crickets against to rule out this alternative.

Please see above and lines 185-189, as well as the new paragraph at line 366, where we have addressed this question in detail and ruled out this scenario.

L. 396: Were regions where π was decreased in the Australian crickets also used? These would imply selection in the Australian crickets.

No. For all of our candidate regions under selection, Australian crickets exhibit significantly higher π than Kauai crickets (lines 696 - 698).

L. 420: it is unclear to me how bootstrapping could remove sampling bias. What kind of sampling bias are the authors referring to here?

We adjusted the language to be more precise. Line 355 now reads:

“To exclude the possibility that stochastic variation during field sampling affected our results ...”

Fig5b: "Precentage" should be "Percentage"

Done (Fig. 5b).

Fig. 5c: gene ratio not ratio

Done (Fig. 5c).

Fig. 5d: It is very unclear to me what the message is here. Is this showing that the two tissue types differ (L487)? But how could I see the tissues here? What does D4, D7, Fw, Nw etc mean? Abbreviations should be explained.

We agree that this panel was difficult to understand. Hence, we replaced it with a panel summarizing the designs of our RNA-seq experiments (Fig. 5d). See response above for details.

Fig. 4b: It seems as if the x axis in the left panels was unlabelled. The label bootstrap value should be moved to the middle.

Done (Fig. 4b).

L. 490: What does "synthetically analysed" mean here? This needs more explanation.

We now define this at its first usage on lines 319 - 320 and on lines 395 - 397.

L. 514-515: turn the parts of the sentences around. First saying what you found and then what you not found.

The first clause has been removed for streamlining (line 433).

L. 506: What does "an increasing number of genes" mean?

Edited to indicate the number of differentiated genes increased over time (line 425-426).

Fig. 6a: It is very difficult to judge if these F_{ST} peaks are indeed higher than expected. Perhaps the horizontal lines show the mean or is it a threshold of 5% top quantile? They need explanation.

Thank you very much for catching this. This was our oversight. The grey horizontal line represents the mean value, and the red line indicates the threshold of the top 5%. Now indicated in the figure legend at line 1375.

Fig. 6: How were these regions chosen?

These regions are the flanking regions of key candidate genes as described on lines 1371-1372. The selection of key candidate genes was based on a combination of our bottom-up and top-down approaches, which considered "gene expression data," "genomic regions differentiated over time," "genomic regions under selection," and "empirical observations in the Kauai population" (see Supplementary Figure 1). Although only one gene was used to represent the pattern of each time interval, all genes described in our manuscript were statistically significant.

Fig. 6ii) Should the F_{ST} here really be between all Hawaiian crickets and the Australian ones? It seems that using only the individuals of the later time point of the comparison should be compared to the Australian crickets as we would expect selection to only start after the transition (e.g. not to be present in 2012).

Thank you very much for your suggestion. We have re-generated these panels accordingly. For Fig.6b and 6c, our analyses were conducted between the later time points and the Australian crickets. For Fig.6a, all Hawaiian crickets were used because this selection pressure started before 2012 (Fig. 1). These new panels consistently corroborated our initial findings and conclusions.

Fig.6 Top candidate genomic regions and phenotypes coevolving with flatwing. Genomic signatures of selection and expression profile for: **a.** the *caiap* homologue on Chromosome 10, which is associated with response to parasitoid fly infestation, **b.** the *RGRF2* homologue on Chromosome 5, which is associated with behavioural plasticity in response to reduced acoustic signals, **c.** the *OPTN* homologue on Chromosome 5, associated with altered mating tactics caused by the absence of singing males. The position of candidate genes is indicated at the top of each panel, accompanied by dashed vertical lines. The flanking regions shown for each candidate gene are proportional to the length of the respective gene. F_{ST} and π were calculated using sliding-window analyses with 10 kb windows to make comparisons (i) between Kauai individuals collected from different years, and (ii) between Kauai individuals of corresponding time points and the Australian control group. Horizontal grey dashed lines denote the mean whole-genome value for the displayed statistics in each panel, while dark red horizontal dashed lines indicate the top 5% threshold values.

L. 556: this information is repetitive as it was already shown in the Introduction and in the previous section.

This sentence has been deleted in the revision (line 437).

L. 603: "seven genes under selection" is a very strong formulation. I would call them candidates for selection. Just a simple F_{ST} difference does not necessarily mean that they really are under selection as it could also be an inversion and some drift or a low-recombination region experiencing strong background selection.

We have now employed a more rigorous expression and referred to these genes as "candidate genes under selection" (e.g. lines:313, 320, 370, 380, 484 etc.). We have thoroughly reviewed our manuscript and ensured that all similar expressions are revised accordingly. Regarding the reliability of our selection analyses, please refer to our responses to your previous questions (see above).

L. 604: What does a difference between Kauai vs Mangaia imply? What is Mangaia? Once one has read the methods and reread the results, it becomes clear that these are the Cook island populations that are mentioned on L499 but one should not have to do such detective work to get this key information. In L499 it does not become evident that a Cook Island population was used for the differential gene expression.

Thank you for your suggestions. We have addressed this issue accordingly. Please refer to our responses to your previous question for more details.

L. 646: citation for the disagreement?

Done (line: 533).

L. 687: citation for the original assembly? What is the quality of this genome?

We have added a citation for the original assembly (line 137, 141). The assembly reported in this study represents a significant improvement for this species. Quality metrics for the new assembly are given in Supplementary Table 1, and comparison of quality metrics for the two *T. oceanicus* genome assemblies is given in Supplementary Table 2. Supplementary Figure 1 visually illustrates the new metrics. All of these were noted starting at line 130 in the manuscript.

L. 717: Should be supplementary data 2, not 1. The abbreviations are impossible to understand.

Thank you for catching this. We have corrected it. We have regenerated Supplementary Data 2 and created two additional figures (Fig. 5d and Supplementary Figure 1) to enhance clarity and facilitate comprehension.

L. 716: It is not clear here that this data was published previously. Please provide citations.

Done (line: 604).

L. 754: Make it clear here which samples are new.

We have noted which samples are new in Supplementary Data 1.

L. 774-448: No sequencing depth filter? Are homozygous sites missing? If yes, how could pi be calculated???

Sorry for the confusion caused by our wording. We have applied a sequencing depth filter, and there are no missing homozygous sites. The same protocol used in our previous work¹ was reused here. We have elaborated on these details in our revised manuscript at lines 664-674.

“Given the XX/XO (female/male) sex determination system observed in crickets, all male samples were assigned homozygous genotypes for X-linked loci^{1,2}. Additionally, a Perl script was employed to filter low-quality SNPs potentially arising from sequencing errors, whereby SNPs with Phred-scaled quality scores < 30 or exhibiting abnormal depth (below one-third or exceeding three times the mean sequencing depth per individual) were filtered out^{1,3,4}. For SNPs located in the X chromosome, criteria for abnormal depth were adjusted to half those applied to autosomes. Heterozygous calls for X-linked SNPs were rectified to homozygous states in cases where one allele exhibited a depth threefold or higher than the other. In cases where neither allele was supported by the supermajority, the genotype was deemed unknown. Loci exhibiting unknown genotypes in over 25% of all individuals were excluded¹. To mitigate potential biases caused by outgroups, SNPs were called using different subsets of individuals and subsequently utilized for subsequent analyses¹.”

References

- 1 Zhang, X., Rayner, J. G., Blaxter, M. & Bailey, N. W. Rapid parallel adaptation despite gene flow in silent crickets. *Nat Commun* 12, 50 (2021).
- 2 Kim, K. W. *et al.* Genetics and evidence for balancing selection of a sex-linked colour polymorphism in a songbird. *Nat Commun* 10, 1852 (2019).
- 3 Qiu, Q. *et al.* Yak whole-genome resequencing reveals domestication signatures and prehistoric population expansions. *Nat Commun* 6 (2015).
- 4 Wang, Z. *et al.* Chromosome-level genome assembly and population genomics of *Robinia pseudoacacia* reveal the genetic basis for its wide cultivation. *Commun Biol* 6, 797 (2023).

L. 777: What does "fixed the heterozygous calls of X-linked SNPs" mean?

Sorry for the confusion caused by our wording. In this revised version, all details have been clarified at lines 664-674 (see above).

REVIEWERS' COMMENTS

Reviewer #1 (Remarks to the Author):

The authors did a nice job addressing the comments raised in the previous review. I think this is a unique study that provides insight into compensatory evolution, especially given it is a natural empirical system. The changes made have improved the clarity of the approaches undertaken and strengthened conclusions. I have no recommended revisions.

Reviewer #2 (Remarks to the Author):

The authors have implemented my comments well. I am particularly pleased to see that the haplotype-based selection statistics support the signatures of selection likely associated with the change in the social environment the silent crickets experienced. I very much like the combination of the bottom-up and top-down approaches to identifying candidate genes. However, I think how these were combined to define a set of candidate genes is in many places a bit unclear. I would thus recommend textual changes to clarify this. It is also still unclear to me how the bootstrapping was performed and how the differential gene expression experiments were combined. I have provided more details on these points and added additional tiny comments below.

Bootstrapping (L356ff, 703ff, Fig. 5b):

I understand better now what and why the authors did here to test the robustness of their selection tests. However, this does not seem like bootstrapping to me if I understand it correctly. Bootstrapping would involve resampling from the dataset with! replacement until the same size is reached. However, what the authors seem to have done here is a leave-one-out approach as they subsampled 9 of 10 individuals without replacement. It is not described how the "bootstrap value" was obtained. A "bootstrap value" in a phylogeny is the percentage of phylogenies from bootstrapped datasets that support a specific node. This is standard and does not need to be described. However, in the genome scan approach, it is unclear what the authors mean with "bootstrap value". This needs to be described as it is not a standard approach. Is it perhaps the proportion of resampling iterations giving a value in the top 5%? How are F_{ST} and π values combined?

It seems to me that there are different ways putative candidate genes under selection were defined throughout the text but it is not always clear what kind the authors refer to, e.g. in Figure 5. Some small textual changes would help a lot here. The Figure 5 legend in particular would benefit from clarification to ensure the readers know what the authors are referring to:

In Figure 5: "How are putatively selected genes" defined? Is this just based on F_{ST}/π or also based on differential gene expression? It seems that this refers only to F_{ST}/π analyses (L328). In the methods (L698) it says that genomic regions containing the top 5% F_{ST} values and top 5% π log ratios were identified as candidate regions. Are "putatively selected genes" genes in these regions if they overlap with at least one window? In the new version (L720) I can see that "a candidate gene" was considered validated if it overlapped with a phased-genotype-based selection signature. L380 shows that 74% of the "original candidate genes" were validated. Does "putatively selected genes" refer to validated genes? How does this gene set differ from the "Kauai-selected gene set" (L336)? Is bootstrapping considered?

Also on L 350 "We identified 525 genomic regions differentiated over time" add what stats were used and how to infer that they are "differentiated" (top5% in both F_{ST} and π log-ratios?). Then in Fig. 5c and L 358: were the GO annotations done based on genes in the top 1% threshold of the bootstrapping? As GO enrichment analyses were performed with the differentially expressed genes as well, it needs to be clarified in the figure legend what GO analysis authors are referring to in Fig. 5c.

On L774 the authors speak of the "official gene set" and on L780 "Genes overlapping the corresponding candidate regions were defined as candidate genes". Is this what the authors refer to in Fig. 5c? So not the same set of candidate genes as in Fig. 5a and b? Again, a clarification in the figure legend would be good. If it is indeed an intersection of genes showing high differentiation and having been shown to be differentially expressed, the figure panel showing the

design of the gene expression analyses should go in front of the GO enrichment analysis. On L. 340 the authors write that GO enrichment analyses based on differentially expressed genes “were used to summarize the biological function of the Kauai-selected gene sets” but “Kauai-selected gene sets” seem to also include genes that are not differentially expressed or do I misunderstand this? Do you mean the intersection of the Kauai-selected gene sets identified with π /Fst and the genes with differential expression?

The new figure panel summarising the RNA experimental design nicely (Fig. 5d) shows that there are two experiments. However, it remains unclear to me how these two experiments were combined to inform which genes are “candidate genes” in the bottom-up approach.

L411: Changing the title from “Synthetic analysis of putative selected genes” to something like “Intersecting the list of candidate genes from the bottom-up and top-down approach” would be much more informative. I am still not really sure what the authors mean with “synthetic analysis” and this term also does not show up in the methods.

Figure 2: I would recommend to highlight that scaffold 1 is the X chromosome, at least mention it in the figure caption.

Very minor point: The superscript numbers indicating citations sometimes are glued to the last word, sometimes are separated by a blank from the last word.

Below we describe thorough, targeted edits to remove any residual confusion about “bootstrapping” (now renamed and explained further) and the connection between gene expression experiments.

REVIEWERS' COMMENTS

Reviewer #1 (Remarks to the Author):

The authors did a nice job addressing the comments raised in the previous review. I think this is a unique study that provides insight into compensatory evolution, especially given it is a natural empirical system. The changes made have improved the clarity of the approaches undertaken and strengthened conclusions. I have no recommended revisions.

We are grateful for the reviewer’s enthusiasm.

Reviewer #2 (Remarks to the Author):

The authors have implemented my comments well. I am particularly pleased to see that the haplotype-based selection statistics support the signatures of selection likely associated with the change in the social environment the silent crickets experienced.

We are grateful for the positive assessment.

I very much like the combination of the bottom-up and top-down approaches to identifying candidate genes. However, I think how these were combined to define a set of candidate genes is in many places a bit unclear. I would thus recommend textual changes to clarify this.

Clarified – see below.

It is also still unclear to me how the bootstrapping was performed and how the differential gene expression experiments were combined. I have provided more details on these points and added additional tiny comments below.

We fixed these outstanding minor points of confusion – see below.

Bootstrapping (L356ff, 703ff, Fig. 5b):

I understand better now what and why the authors did here to test the robustness of their selection tests. However, this does not seem like bootstrapping to me if I understand it correctly. Bootstrapping would involve resampling from the dataset with! replacement until the same size is reached. However, what the authors seem to have done here is a leave-one-out approach as they subsampled 9 of 10 individuals without replacement. It is not described how the “bootstrap value” was obtained. A “bootstrap value” in a phylogeny is the percentage of phylogenies from bootstrapped datasets that support a specific node. This is standard and does not need to be described. However, in the genome scan approach, it is unclear what the authors mean with “bootstrap value”. This needs to be described as it is not a standard approach.

We replaced the term “bootstrap” with “resampling procedure”. Resampling describes a broader category of statistical procedures, and is both more technically accurate and enables us to explain our approach in greater detail (lines 366, 708-727, 1363, 1364, 1367).

*“We performed a validation procedure using iterative resampling 100 times for each differentiation analysis. Specifically, we adopted a subsampling strategy in which 9 individuals were randomly selected without replacement from the population of 10 individuals at each time point. This step yielded a total of 10 scenarios for each time-divided group, resulting in 30 sub-sampled populations (10 scenarios * 3 time points). The differentiation analysis described in the paragraph above was then performed iteratively using these sub-sampled populations. The outcomes of these analyses were used to calculate resampling statistics for each candidate gene significantly differentiated between time points; this statistic reports the percentage of resampling iterations that support the candidate gene in question. This approach enabled us to obtain a more reliable estimate of significance for observed differentiation and selection signals. The threshold for resampling statistical significance was determined to be the top 1% of the empirical distribution of resampling statistics across the whole genome.”*

Is it perhaps the proportion of resampling iterations giving a value in the top 5%?

The threshold for resampling significance is the top 1% (lines 726-727).

How are F_{ST} and π values combined?

We clarify our sliding-window method at lines 356 and 357. It calculated π log-ratios and F_{ST} values for each window. The windows containing both the top 5% of F_{ST} values and the top 5% of π log-ratios were identified as candidate selected regions.

It seems to me that there are different ways putative candidate genes under selection were defined throughout the text but it is not always clear what kind the authors refer to, e.g. in Figure 5. Some small

textual changes would help a lot here. The Figure 5 legend in particular would benefit from clarification to ensure the readers know what the authors are referring to:

Only one method was used to define “candidate genes under selection / genes differentiated over time”, which we clarify at lines 347-348 and 702-705. See further details below. Other methods were either used to validate the robustness of our original selection / differentiation analyses (line 715-727) or to highlight the biological function of these sets of “candidate genes under selection” (line 788-795).

In Figure 5: “How are putatively selected genes” defined? Is this just based on F_{ST}/π or also based on differential gene expression? It seems that this refers only to F_{ST}/π analyses (L328).

This gene set is detected by comparing π log-ratios and the population differentiation F_{ST} index as above. Caption clarified at lines 1362-1364. [Fig. 6 in our revised manuscript is the Fig. 5 of our original manuscript, see below.]

In the methods (L698) it says that genomic regions containing the top 5% F_{ST} values and top 5% π log ratios were identified as candidate regions. Are “putatively selected genes” genes in these regions if they overlap with at least one window?

Thanks for raising this, it was confusing as written. We clarify:

- (a) **What a *candidate selected region* is (line 702)**
- (b) **That there are two types of *candidate selected regions*: one type is “*Kauai-selected regions*” and the other is “*regions differentiated over time*” (lines 703-705)**
- (c) **That these regions were defined using the F_{ST}/π -log-ratio sliding window approach (line 705-710)**
- (d) **That genes that overlap at least one such candidate region are designated as candidate genes (line 711)**
- (e) **That these candidate selected genes correspond to “*Kauai selected genes*” and “*genes differentiated over time*”, respectively (lines 712-714).**

In the new version (L720) I can see that “a candidate gene” was considered validated if it overlapped with a phased-genotype-based selection signature. L380 shows that 74% of the “original candidate genes” were validated. Does “putatively selected genes” refer to validated genes?

No, this refers to the original candidate genes. We changed this to: “Of all candidate genes (*Kauai-selected genes and genes differentiated over time*) detected using our bottom-up approach (Supplementary Figure 1), 74% were further validated using these phased haplotype approaches (Supplementary Table 10).” (lines 390-392) to avoid confusion.

How does this gene set differ from the “*Kauai-selected gene set*” (L336)? Is bootstrapping considered? Also on L 350 “We identified 525 genomic regions differentiated over time” add what stats were used and how to infer that they are “differentiated” (top5% in both F_{ST} and π log-ratios?).

We clarify that this refers to genes that appeared in one, the other, or both sets of candidates (*Kauai-selected genes* and *genes differentiated over time*) (lines 390-392). And yes, they represent the top 5% -- this is in the Methods but we now note it at lines 356-357.

Then in Fig. 5c and L 358: were the GO annotations done based on genes in the top 1% threshold of the bootstrapping? As GO enrichment analyses were performed with the differentially expressed genes as well, it needs to be clarified in the figure legend what GO analysis authors are referring to in Fig. 5c.

The enrichment was done on “*genes differentiated over time*” – clarified at line 368 and lines 1363 -1364, 1369 in the figure legend.

On L774 the authors speak of the “official gene set” and on L780 “Genes overlapping the corresponding candidate regions were defined as candidate genes”. Is this what the authors refer to in Fig. 5c? So not the same set of candidate genes as in Fig. 5a and b? Again, a clarification in the figure legend would be good.

“Official gene set” referred to the entirety of genes predicted in the whole genome but we can see how this gets confusing. We changed the terminology to “official reference gene list” at lines 629, 633, 769, 770, 788, 794.

“Genes overlapping the corresponding candidate regions were defined as candidate genes” was a general description of how candidate genes were identified in our study. Clarified at lines 793 -795.

If it is indeed an intersection of genes showing high differentiation and having been shown to be differentially expressed, the figure panel showing the design of the gene expression analyses should go in front of the GO enrichment analysis.

No, the GO enrichment analyses (Fig. 6c) were performed based only on “*genes differentiated over time*”. We rephrased the legend to avoid misunderstanding (line 1369).

We also split Fig. 5 into two figures in response to the editor's request to shorten the figure legends to 350 words.

On L. 340 the authors write that GO enrichment analyses based on differentially expressed genes “were used to summarize the biological function of the Kauai-selected gene sets” but “Kauai-selected gene sets” seem to also include genes that are not differentially expressed or do I misunderstand this? Do you mean the intersection of the Kauai-selected gene sets identified with pi/Fst and the genes with differential expression?

No. This is a functional enrichment analysis of the candidate gene set before any intersection with DEGs. To avoid misunderstanding, we revised these two sentences: “*These candidate genomic regions, and sets of genes located in these regions, are referred to as “Kauai-selected regions” and “Kauai-selected genes”, respectively. Gene Ontology (GO) enrichment analyses were used to summarize the biological function of this gene set*” (lines 347-349).

The new figure panel summarising the RNA experimental design nicely (Fig. 5d) shows that there are two experiments. However, it remains unclear to me how these two experiments were combined to inform which genes are “candidate genes” in the bottom-up approach.

We clarify at line 806-808 that the results obtained from these two RNA-seq experiments (i.e. the three DEG sets described in Fig. 5 [Previous Fig. 5d]) were used independently to overlap with the “*Kauai-selected genes*” and the “*genes differentiated over time*”.

L411: Changing the title from “Synthetic analysis of putative selected genes” to something like “Intersecting the list of candidate genes from the bottom-up and top-down approach” would be much more informative. I am still not really sure what the authors mean with “synthetic analysis” and this term also does not show up in the methods.

We sympathise with the reviewer. A few simple edits now clarify what the synthetic analysis is and how it identifies regions undergoing compensatory coevolution:

- (a) We spelled out that the synthetic analysis is the overlapping of our bottom-up and top-down approaches (lines 409-412, 416-417, 800-801, 805-808).
- (b) We describe this enrichment test and the biological logic (line 419-421).
- (c) We changed the subheading of the Results section at 414 to be more declarative and say exactly what the analytical outcome is: “Synthetic analysis to identify genes implicated in compensatory intragenomic coadaptation”
- (d) We now identify the term ‘synthetic analysis’ in the relevant Methods subheading (line 787) and in a new paragraph at line 800.
- (e) We edited Fig. 6 and Fig. 7 titles to indicate that what is shown are genome-wide patterns of compensatory coevolution and top regions/genes involved in compensatory genomic coevolution, respectively (lines 1360 and 1411)

Figure 2: I would recommend to highlight that scaffold 1 is the X chromosome, at least mention it in the figure caption.

Done (line 1237).

Very minor point: The superscript numbers indicating citations sometimes are glued to the last word, sometimes are separated by a blank from the last word.

Corrected.